# Ice viscosity governs hydraulic fracture causing rapid drainage of supraglacial lakes

Tim Hageman[a], Jessica Mejía[b], Ravindra Duddu[c,1], and Emilio Martínez-Pañeda[a,d,2]

[a]Department of Engineering Science, University of Oxford, Oxford OX1 3PJ, UK
[b]Department of Geology, University at Buffalo, Buffalo, NY 14260, USA
[c]Department of Civil and Environmental Engineering, Department of Earth and Environmental Sciences, Vanderbilt University, Nashville, TN 37235, USA
[d]Department of Civil and Environmental Engineering, Imperial College London, London SW7 2AZ, UK

**Correspondence:** Ravindra Duddu (ravindra.duddu@vanderbilt.edu) and Emilio Martínez-Pañeda (emilio.martinez-paneda@eng.ox.ac.uk)

**Abstract.** Full thickness crevasses can transport water from the glacier surface to the bedrock where high water pressures can open kilometre-long cracks along the basal interface, which can accelerate glacier flow. We present a first computational modelling study that describes time-dependent fracture propagation in an idealised glacier causing rapid supraglacial lake drainage. A novel two-scale numerical method is developed to capture the elastic and viscoelastic deformations of ice along with crevasse propagation. The fluid-conserving thermo-hydro-mechanical model incorporates turbulent fluid flow and accounts for melting/refreezing in fractures. Applying this model to observational data from a 2008 rapid lake drainage event indicates that viscous deformation exerts a much stronger control on hydrofracture propagation compared to thermal effects. This finding contradicts the conventional assumption that elastic deformation is adequate to describe fracture propagation in glaciers over short timescales (minutes to several hours) and instead demonstrates that viscous deformation must be considered to reproduce observations of lake drainage rate and local ice surface elevation change. As supraglacial lakes continue expanding inland and as Greenland Ice Sheet temperatures become warmer than -8°C, our results suggest rapid lake drainages are likely to occur without refreezing, which has implications for the rate of sea level rise.

## 1 Introduction

Mass loss through melting and iceberg calving from the Greenland Ice Sheet (GrIS) will continue increasing throughout the century in response to atmospheric warming (Bamber et al., 2019; Bevis et al., 2019; Pattyn, 2018). Under the highest baseline emissions scenario RCP8.5, mass loss from the GrIS could raise global sea level by up to 90±50 mm by 2100 (Goelzer et al., 2020). While this mass loss is dominated by increased runoff derived from surficial melting (Hofer et al., 2020), the influence of meltwater increase on ice dynamics remains unresolved and unaccounted for in ice sheet models. To better parameterise the coupling between melting and ice dynamics in ice sheet models it is essential to develop and utilise advanced physics-based models to better understand the processes at the smaller (glacier) scale.

When crevasses reach the ice sheet's base they can transfer liquid water from the surface to the base of ice sheets, where it can influence ice dynamics by modulating pressures within the subglacial drainage system. Meltwater produced on the surface

of glaciers and ice sheets often collects in depressions forming supraglacial lakes that can drain rapidly when intersected by a crevasse (Das et al., 2008; Doyle et al., 2013; Chudley et al., 2019; Selmes et al., 2011; Smith et al., 2015; Christoffersen et al., 2018). When water-filled, crevasses can continue to propagate deeper in the ice sheet until reaching the bed (van der Veen, 2007; Weertman, 1973), thereby creating direct connections between the supraglacial and subglacial drainage systems. It has been observed that only a limited amount of water is required to create this connection (Krawczynski et al., 2009; Selmes et al., 2011). However, when these hydraulically-driven fractures connect to supraglacial lakes, they have the ability to transport massive amounts of water to the ice sheet's base (Lai et al., 2021). These large volumes of water quickly overwhelm and pressurise the subglacial drainage system to initiate horizontal basal fracture propagation and uplift, changes in basal friction, and acceleration of the overlying ice (Das et al., 2008; Stevens et al., 2015; Doyle et al., 2013; Liang et al., 2012; Chudley et al., 2019; Shannon et al., 2013) that can extend tens of kilometres down-glacier (Andrews et al., 2018; Hoffman et al., 2011; Mejía et al., 2021).

Crevasses are usually modelled as opening (mode I) fractures penetrating through the ice thickness under the action of tensile stress (van der Veen, 2007). Most existing models for estimating the depth of water-filled crevasses rely on the assumption that the flow of water associated with fracture propagation is such that hydrostatic conditions are valid and the thermal process is slow enough that melting and freezing can be neglected. Based on this assumption, analytical linear elastic fracture mechanics (LEFM) based models were used to estimate the depth of water-filled crevasses, namely, the Nye zero stress (Jezek, 1984; Benn et al., 2007; Nick et al., 2010), dislocation-based LEFM (Weertman, 1971), and stress-intensity-factor-based LEFM models (Smith, 1976; Van Der Veen, 1998). Recently, stress-intensity-factor-based LEFM models for estimating water-filled crevasse depth have been proposed using finite-element-based (Jimenez and Duddu, 2018) and boundary-element-based methods (Zarrinderakht et al., 2022). Alternatively, continuum damage mechanics and phase field models for water-filled crevasse propagation have been more recently developed (Duddu et al., 2020; Clayton et al., 2022). In the above-mentioned studies, except for Zarrinderakht et al. (2022), the water column height within the crevasse was prescribed to drive crevasse propagation rather than the volume of water.

The mechanical response of glacier ice is usually described by the isotropic linear elastic model over short time scales (micro-seconds to a few minutes) for simplicity, but over longer time scales (hours to months) ice response is better described by the nonlinear viscoelastic model (i.e., dependent on the strain-rate, temperature, and time), known as Glen's law (Glen, 1955). A few studies in the literature incorporated viscous deformation of ice and investigated longer-term processes such as the slow propagation of crevasses over weeks to months (Poinar et al., 2017) or moulin formation post-hydrofracture (Andrews et al., 2022). Other studies described hydraulically driven (vertical) crevasse propagation that subsequently leads to horizontal fracture propagation at the ice-bedrock interface and ice sheet uplift using approximate analytic solutions (Tsai and Rice, 2010, 2012), or using large-scale mass and momentum balances (Rice et al., 2015; Andrews et al., 2022). Within numerical simulations, reduced-order models have been used to include the interactions between basal water and basal friction (Pimentel and Flowers, 2011). Some studies utilised the finite element method to accurately simulate the deformations of the ice sheet including creep (Hewitt et al., 2012; De Fleurian et al., 2014; Crawford et al., 2021). However, there exists no comprehensive

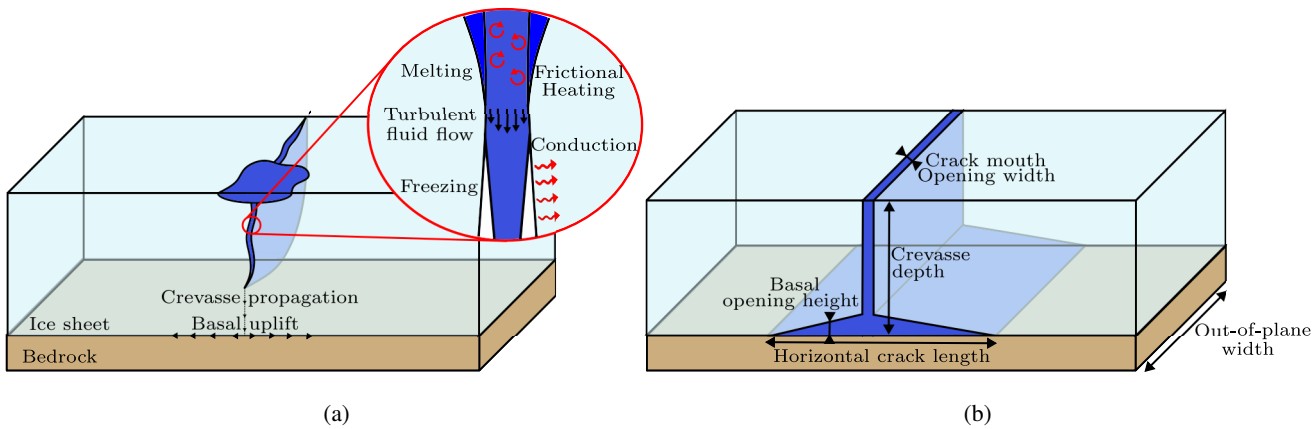

**Figure 1.** Schematic diagrams describing the physical phenomena associated with lake drainage driven by hydraulic fracture. (a) In the 3D real case scenario, the vertical crevasse below the lake propagates and eventually reaches the bedrock, upon which fluid flow is sustained by horizontal fracture propagation and basal uplift. (b) Assuming uniformity in the out of plane direction, the problem can be idealised assuming 2D plane strain conditions to reduce the computational burden.

numerical model capable of describing both the vertical and horizontal fracturing and uplifting due to the turbulent flow of pressurised meltwater within the fracture along with melting and refreezing.

In this work, we present a comprehensive numerical model for hydraulic fracturing that takes into account elastic-viscoplastic deformation and thermal processes driven by turbulent water flow (see Methods for full details). We use this model to investigate how the choice of a viscoelastic or linear-elastic ice sheet rheology influences fracture propagation, thermal processes (refreezing and frictional heating within crevasses), and ice sheet uplift. We also examine the requirements for fluid supply from supraglacial lakes and the timescales involved during the fracturing process by comparing model outputs with observations from rapid lake drainage on the Greenland Ice Sheet. Ultimately, our results highlight the important role of creep deformation in facilitating rapid supraglacial lake drainage events thereby supporting the incorporation of a viscoelastic ice rheology even on short (hours) timescales, whereas the effects of melting were less relevant at these timescales.

## 2   Models and Methods

To investigate the hydraulic fracturing process induced by supraglacial lakes, the 3D geometry from Fig. 1a is simplified by assuming crevasse opening is uniform over the full out-of-plane width (see Fig. 1b). This allows the domain to be approximated as a 2D domain, shown in Fig. 2. Here, we note that this assumption requires the out-of-plane length of the crevasse to be large, such that basal uplift is produced by the uni-directional fluid flow away from the vertical crevasse. An alternative assumption to reduce the problem from three-dimensions to two-dimensions would be to consider the crevasse as a vertical cylindrical conduit to allow for radial subglacial water flow. Using an axisymmetric representation would be appropriate for crevasses with surface lengths (i.e., the out-of-plane direction from Fig. 1b) much shorter compared to the length of the horizontal/radial basal crack, such that the vertical crevasse can be considered as a conduit propagating downwards. In contrast, the plane-strain

representation used here is suitable for when the surface crevasse spans longer distances, such that the crack is considered as a plane propagating downwards. While our 2D model for horizontal basal crack propagation and basal uplift is valid for the axisymmetric assumption, it would not be appropriate to assume that the vertical crevasse is radially symmetric because it would no longer represent a planar fracture undergoing opening, but rather the creation of a cylindrical conduit. This is an important distinction because of the associated processes governing fracture propagation. For a typical planar or "vertical" crevasse formed under plane strain, mechanical stresses drive crevasse propagation. In contrast, under axisymmetric conditions, the cylindrical conduit would propagate downwards due to fracture, melting and erosion processes. While it is possible to model it as a cylindrical moulin evolution (Trunz et al., 2022), but it is difficult to describe the fracture mechanics using the existing framework. Of course a 3D model able to capture both these phenomena would be ideal, but it would be computationally too expensive, so we utilise the 2D plane strain approximation, focusing on the propagation of fractures driven by stresses and only consider the lateral melting of the fracture faces to ultimately align with the observed morphology of crevasses..

It is further assumed that crevasse length is much larger than its width, defined as the spacing between crevasse walls measured at the ice surface. This assumption allows the separation of two distinct spatial scales: the glacier scale on which deformations occur, and the fracture scale which determines the fluid flow and thermal effects (melting or refreezing, frictional heating, and conduction). The 2D domain consists of a $6\,\mathrm{km} \times 980\,\mathrm{m}$ portion of an ice sheet on a $200\,\mathrm{m}$ thick rock layer. The thickness of the rock layer is chosen to be large enough so that the stresses around the basal crevasse are not spuriously altered by the zero vertical displacement boundary condition applied to the bottom surface of the rock layer. An initial $30\,\mathrm{m}$ deep crevasse connected to a supraglacial lake is assumed to be present at the ice surface. Given the density difference between the water and surrounding ice, a 30-m crevasse depth is sufficient to overcome the tensile strength of the ice and initialise the hydraulically driven crevasse propagation.

The domain geometry is a 2D idealisation of a lake drainage event recorded in Das et al. (2008), utilising the same ice thickness and ice material properties (see Table 1). To enforce a no-slip basal boundary condition during crevasse propagation we assume the ice-bedrock interface is frozen. Although the ice-bedrock interface is not frozen in the North Lake area of Greenland modelled here, we implement the frozen-bed assumption to isolate hydraulically induced basal crevasse growth. A full discussion of this rationale is provided in Section 2.1.2. A further simplification made is that no defects exist within the ice, and the rock layer is perfectly impermeable, preventing any fluid leak-off from the base of the ice sheet. Imposing no leak-off from the water-filled crevasse deviates from realistic conditions, where water may slowly seep into the bedrock layer at the base. This leak-off will (slightly) reduce the water pressure within the crevasse while the surrounding bedrock is saturated, slightly slowing down the propagation of the horizontal crack. While we acknowledge that these effects might play a significant role under certain circumstances, accurately capturing the full glacial drainage system (e.g. subglacial drainage channels) is beyond the scope of this study. In the discussion section we comment on the potential implications of this simplification.

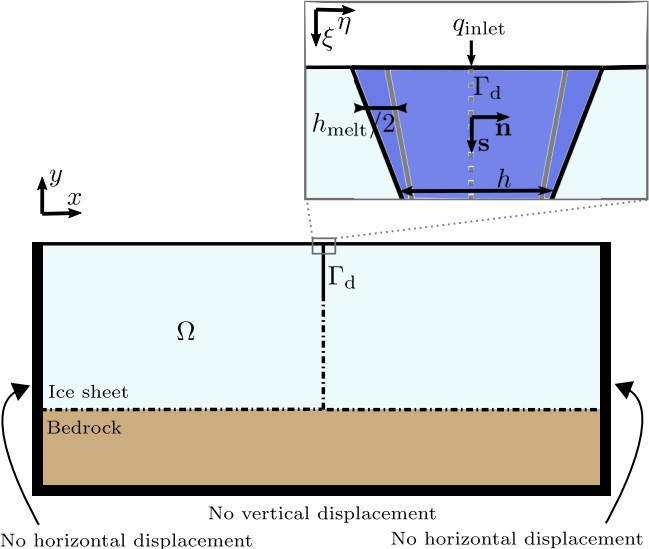

**Figure 2.** Schematic sketch showing the macro-scale domain description, boundary conditions, and the in-plane $x - y$ coordinates assuming 2D plane strain conditions. The ice and rock domain is denoted by $\Omega$ and the fracture interface is denoted by $\Gamma_d$. The inset image shows the micro-scale domain and the corresponding $\xi - \eta$ coordinates used for describing turbulent fluid flow and thermal conduction.

A finite element formulation is used to accurately capture the fracturing and subsequent uplift process. This formulation includes the linear elastic and nonlinear viscoelastic deformations of the ice [1]. This allows for the inclusion of both shorter-term (seconds to minutes) elastic deformations and longer-term (hours to months) viscous relaxation. As the nonlinear viscous response is dependent on the magnitude of the deviatoric stresses, its effect becomes relevant in areas with rapidly changing stress states (i.e., crack tips) even on shorter timescales. We explicitly include the conservation of mass within the crack to track the fluid flow as it moves through the crevasse. In addition, thermal processes (melting, frictional heating, and thermal diffusion) are included by means of a new small-scale formulation. In the remainder of this section, these different components will be discussed in more detail.

## 2.1 Momentum balance and constitutive models

The ice sheet through which hydrofracture occurs is modelled as the domain $\Omega$ in Fig. 2. This domain uses the plane-strain assumption to simplify the three-dimensional geometry from Fig. 1b to the two-dimensional geometry from Fig. 2. The domain is composed of a layer of ice resting on a layer of deformable rock, which are described through their displacements $\mathbf{u}$ and the history-dependent viscous strains $\boldsymbol{\varepsilon}_v$. Within this domain a discontinuity is present, $\Gamma_d$, which represents both the vertical crevasse and the horizontal basal cracks (in both directions away from the vertical crevasse). The current condition of this

---

[1]The term "viscoelastic" is used to refer to the combination of reversible elastic deformations and irreversible viscous/plastic deformations resulting from the use of Hooke's and Glen's laws. While this terminology is more common in the non-Newtonian fluids/rheology and glaciology community, in the solid mechanics community these models are referred to as viscoplastic models, namely the Norton-Hoff and Bingham-Maxwell models.

fluid-filled fracture is described through its pressure $p$, and its history is included through the time-since-fracture $t_0$ and the thickness of wall melting/refreezing $h_{\text{melt}}$ (positive for melting, negative for freezing).

Hydraulic fracture occurs rapidly over a short time scale. As such, the ice will exhibit both elastic deformations due to sudden fracture propagation, as well as some viscous relaxation with time. To describe these two mechanisms, ice is considered to be a deformable solid material (Duddu and Waisman, 2012), in contrast to the commonly used non-Newtonian viscous liquid for long-term ice flow simulations (Larour et al., 2012; Lipscomb et al., 2019). Within this description, the momentum balance is solved to obtain the displacements $\mathbf{u}$ throughout the domain:

$$\rho_\pi \ddot{\mathbf{u}} - \boldsymbol{L}^T \boldsymbol{\sigma} = \rho_\pi \mathbf{g}, \tag{1}$$

where the density $\rho_\pi$ corresponds to either the bedrock density $\rho_r$ or the ice density $\rho_i$, $\mathbf{g} = [0 \ -9.81 \text{ m/s}^2]^T$ the gravitational acceleration vector, and $\boldsymbol{L}$ is the matrix mapping the displacement to linearised strain vector $\boldsymbol{\varepsilon} = [\varepsilon_{xx} \ \varepsilon_{yy} \ \varepsilon_{zz} \ \varepsilon_{xy}]^T = \boldsymbol{L}\mathbf{u}$. The rock is assumed to be a linear elastic solid, whereas ice is assumed to be a Norton-Hoff type elastic-viscoplastic material. Thus, the stress in ice is defined by the linear-elastic strain, which is obtained by offsetting the total strain with the viscous strain history. The constitutive law for ice and rock is defined by:

$$\boldsymbol{\sigma} = \boldsymbol{D}_\pi \boldsymbol{\varepsilon}_e = \boldsymbol{D}_\pi \left( \boldsymbol{\varepsilon} - \boldsymbol{\varepsilon}_v \right) = \boldsymbol{D}_\pi \boldsymbol{L}\mathbf{u} - \boldsymbol{D}_\pi \boldsymbol{\varepsilon}_v, \tag{2}$$

where the isotropic linear elastic stiffness matrix $\boldsymbol{D}_\pi$ is dependent on the material parameters of ice and rock. The viscous strains $\boldsymbol{\varepsilon}_v$ are obtained by integrating the viscous strain rate obtained from the Glen's law as (Glen, 1955):

$$\begin{aligned}
\dot{\boldsymbol{\varepsilon}}_v &= A \left( \boldsymbol{\sigma}_{\text{dev}}^T \boldsymbol{\sigma}_{\text{dev}} \right)^{\frac{n-1}{2}} \boldsymbol{\sigma}_{\text{dev}} \\
&= A \left( \left( \mathbf{u}^T \boldsymbol{L}^T - \boldsymbol{\varepsilon}_v^T \right) \boldsymbol{D}^T \boldsymbol{P}^T \boldsymbol{P} \boldsymbol{D} \left( \boldsymbol{L}\mathbf{u} - \boldsymbol{\varepsilon}_v \right) \right)^{\frac{n-1}{2}} \\
&\qquad \boldsymbol{P} \boldsymbol{D} \left( \boldsymbol{L}\mathbf{u} - \boldsymbol{\varepsilon}_v \right),
\end{aligned} \tag{3}$$

where the projection matrix $\boldsymbol{P}$ is used to obtain the deviatoric part of the stress vector $\boldsymbol{\sigma}_{\text{dev}} = \boldsymbol{P}\boldsymbol{\sigma}$. The creep coefficient $A$ is set to zero for the rock layer to solely allow for viscous deformation within the ice, and its temperature dependence is described using the Arrhenius law in the material properties section below. We assume temperature changes caused by the water-filled crevasse to be localised close to the crevasse, so the temperature field and the creep coefficient $A$ are time-invariant within the bulk of the ice sheet.

Throughout this work, ice is considered as a viscoelastic solid, which encompasses the special case of linear elastic solid. Within the viscoelastic model, Eqs. (2) and (3) are used to capture both the immediate elastic response, as well as the slower viscous strains occurring over time. In contrast, the linear elastic model neglects the viscous strains, setting $\dot{\boldsymbol{\varepsilon}}_v$ to zero throughout the simulation.

### 2.1.1 Viscous and elastic time-scales

The inclusion of viscous strains introduces a time-scale over which the mechanical behavior of ice changes from compressible linear elastic solid to incompressible visco-plastic solid. For materials described by a linear viscous model, this time-scale

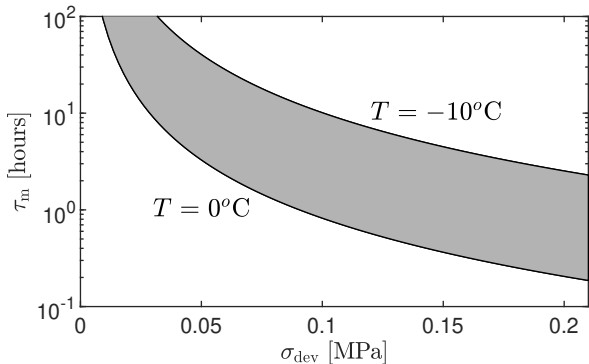

**Figure 3.** Range of Maxwell time-scales following from Eq. (4) and the material parameters from Table 1, assuming an effective deviatoric stress ranging from near zero to the tensile strength of the ice. Shaded area indicates the range of the timescale due to variations of temperature (and thus creep coefficient) with depth. Logarithmic scale used for vertical axis, ranging from 6 minutes to 100 hours.

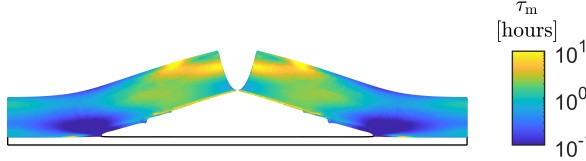

**Figure 4.** Range of Maxwell time-scales following from Eq. (4) and the material parameters from Table 1, evaluated after 2 hours of hydraulic fracture propagation. Deformations are magnified by $\times 1000$, and a logarithmic colour scale is used.

is referred to as the relaxation time (also referred to as Maxwell time-scale), given by (Jellinek and Brill, 1956):

$$\tau_m = \frac{\eta}{\frac{E}{2(1+\nu)}} = \frac{2(1+\nu)}{EA\sigma_{\text{dev}}^{n-1}} \tag{4}$$

where the effective viscosity for a Glen's law viscoelastic material is given by $\eta^{-1} = A\sigma_{\text{dev}}^{n-1}$ (with $A$ being temperature dependent via Eq. (21)).

For the material properties used in this study, see Section 2.4, the range of this time-scale is shown in Fig. 3. The immediate
deformation response of ice is solely determined by the linear elastic strain; however, over a time comparable to the Maxwell time-scale the viscous strains become dominant in driving ice deformation. The rate of increase in creep strain greatly varies depending on the local deviatoric stress. Away from the crack tip, the deviatoric stress is small ($< 0.05$ MPa) so the Maxwell time-scale would be large (on the order of hours to days) implying that elastic strain dominates the deformation response. However, at the crack tip, the deviatoric stress is larger ($\approx 0.21$ MPa), so the Maxwell time-scale would be small (on the order
of minutes) implying that viscoelastic strains can influence the deformation response during hydraulic fracture. This is evident from Fig. 4, where the range of deviatoric stress values during a simulation is used to evaluate the range of Maxwell time-scales existing within the glacier domain at two different ice temperatures.

### 2.1.2 Cohesive fracture model

The force balance at the fracture interface is described by the traction $\tau = \sigma \cdot n$ which is decomposed into cohesive and pressure components as:

$$\tau = \tau_{CZM} + pn \tag{5}$$

where $p$ is the pressure of the water within the crevasse acting normal to the fracture faces. It is assumed that the pressurised fracture propagates solely in mode I (tension-driven fracture). When modelling sharp cracks within a linear elastic material, the stresses at the crack tip become singular. Propagation criteria in linear elastic fracture mechanics are then required to take this singularity into account, and capturing the stress intensity around the crack requires a very fine mesh. However, experimental observations in elasto-plastic materials reveal the existence of a finite fracture process zone or cohesive zone, wherein the stress are bounded by a characteristic material strength. To alleviate these physical and numerical issues with modelling sharp cracks within the finite element method, a cohesive zone model (CZM) is used to regularise the stresses at crack tip region. The CZM is a type of damage mechanics model, in which the traction response of the interface is degraded based on the crack separation or opening width due to damage evolution, and has been widely used to study fracture/delamination of composite materials (Ghosh et al., 2019) and hydraulic fracture in rocks (Hageman and de Borst, 2021; Hageman et al., 2019). Here we use the exponential traction-separation law to define the cohesive traction for mode I cracking as:

$$\tau_{CZM} = -f_t n \exp\left(-[\![u]\!] \cdot n \frac{f_t}{G_c}\right) \tag{6}$$

This cohesive traction depends on the fracture release energy $G_c$, and the tensile strength of the ice $f_t$, unlike linear elastic fracture mechanics that ignores tensile strength. Before cracking, the ice ahead of the crack tip is assumed to be fully intact and undamaged. The crack propagates once the stress within the ice, normal to the prescribed crack direction, exceeds the tensile strength, $\sigma_{yy} > f_t$ for the horizontal crack and $\sigma_{xx} > f_t$ for propagation of the vertical crevasse. As the weight of the ice induces compressive (negative) stresses within the ice-sheet, the total change in stress needed to propagate the crack is therefore given by $(\sigma - \sigma_0) \cdot n > f_t + \rho_i g(H - y)$. For the horizontal crack, this implies that even through a frozen rock-ice interface is assumed with a tensile strength, the majority of the stresses that need to be overcome are a result of the weight of the ice and not the tensile strength. This frozen bed also imposes a no-slip boundary condition between the ice and the bed, requiring displacements in the ice to be matched with the basal rock it is contacting. Upon fracture, this constraint is released and the ice and the rock can move independently in both normal direction (causing crack opening) and tangential direction (causing slip between the ice and rock layers). It should be noted that assuming the bed to be frozen has implications for the downward crevasse propagation and crevasse opening width after the vertical crevasse reaches the base, which is only driven by the elastic and viscous deformations. In contrast, were frictional sliding be allowed at the glacier bed even before the onset of horizontal crack and uplift, an additional opening width would be created due to the two "sides" of the ice-sheet sliding apart. The effect of the glacier sliding induced crevasse widening would be significant if the basal friction is weak.

We use an extrinsic-type CZM implementation, in which the interface elements are inserted dynamically into the finite element mesh as the crack propagates. As a result, the cohesive zone model is only applied post-cracking, while the not-yet-fractured interface behaves as an intact material. This is unlike other conventional intrinsic CZM implementations, wherein the

interface elements are inserted *a priori* ahead of the crack tip along the potential crevasse path, which may result in additional and non-physical displacements around the crack tip (Boone and Ingraffea, 1990). The exponential traction-separation law used here defines a length scale $\ell \approx EG_\mathrm{c}/f_\mathrm{t}^2$ ($\ell \approx 2.3\,\mathrm{m}$ for our case). This length scale gives an indication of the fracture

process zone ahead of the traction-free crack, where the normal traction varies nonlinearly depending on the crack separation or opening width. Taking the length scale $\ell$ close to zero approximates brittle fracture, but requires extremely small interface elements to accurately capture the traction both ahead and behind the crack tip. In contrast, taking larger values of $\ell$ leads to a deviation from fully brittle fracture, instead emulating ductile effects near the crack tip, but allows larger elements to be used while still adequately capturing the stresses and propagation behaviour. Thus, through the choice of an appropriate length scale,

our implementation allows for reasonably sized elements to be used, facilitating the simulation of cracks on glacier-scale in a thermodynamically-consistent and computationally-tractable manner.

## 2.2 Thermo-hydro-mechanical flow model within the fracture

Within the fracture, fluid flow is considered to be driven by gravity and a small inlet pressure imposed at the crevasse mouth (i.e. the top of the domain). This pressure is required to allow the crevasse to open initially, after which the main driving force is

210 the density difference between the glacier ice and water pressurising the fracture. The fluid flow induces heating of the fracture walls due to friction, while the surrounding ice leaches heat from the fluid flow. This behaviour is included through a two-scale scheme: resolving the fluid mass conservation within the fracture at the macro-scale (implemented through a standard finite-element scheme) while resolving the thermal processes as micro-scale effects (numerically resolved on a per-integration-point basis).

### 2.2.1 Pressure-driven flow model

The fluid (meltwater) flow is assumed to be solely contained within the fracture, and both the intact ice and rock are taken as impermeable and non-porous. Thus, we neglect the loss of fluid through fracture walls due to any flow through porous media. The fracture-local coordinate frame $(\xi, \eta) = \mathbf{R}(x, y)$ is defined, using the rotation matrix $\mathbf{R}$ to align the coordinate frame with the fracture direction $\mathbf{s}$ and fracture normal $\mathbf{n}$. A common assumption for flows within fractures is a laminar flow profile,

resulting in a fluid flux given by the cubic law (Witherspoon et al., 1980). While this is accurate for most cases in hydraulic fracturing of rock where typical crack apertures are of the order of millimetres, crevasses have been observed to attain openings of the order of meters. To more accurately model the flow through such large crevasses, it is assumed that the combination of fracture aperture, wall roughness and driving pressure are sufficiently large to cause the flow within the crack to exhibit a turbulent flow profile. The total fluid flux flowing through a fracture with opening $h$ is therefore given through a Gauckler-

Manning-Strickler type flow law (Gauckler, 1867; Strickler, 1981), resulting in a volume flux $q$ produced by turbulent flow as (Tsai and Rice, 2010, 2012):

$$-h\left(\frac{\partial p}{\partial \xi} - \rho_\mathrm{w}\mathbf{g}\cdot\mathbf{s}\right) = \frac{f}{4}\frac{\rho_\mathrm{w}}{h^2}\,|q|\,q, \tag{7}$$

where $\rho_\mathrm{w}$ is the water density and $\mathbf{g}\cdot\mathbf{s}$ is the gravity component acting on the fluid. By comparing the fluid flux within the crevasse during the simulations, it has been verified that assuming a turbulent flow is a good approximation. For instance, the

fluid flux near the inlet stabilises around $1000 \text{ m}^3/\text{m/h}$ (see results section and Fig. 12), providing a Reynolds number for this flow as $Re = \rho_\text{w} |q|/\mu_\text{w} \approx 3 \cdot 10^5$, well above the typical transition point from laminar to turbulent flow ($Re > \mathcal{O}(10^3)$). We note that, while this fluid flux is indeed turbulent in the majority of the crevasse, it is unlikely to be close to the crack tips. For simplicity, we use this turbulent relation throughout the crevasse, however, slightly more realistic results could potentially be obtained by dynamically switching between laminar and turbulent flow models based on the current fluid flux. Although we do not present any results here for other flow models, simulations using a laminar flow model have also been performed (with the fluid flux scaling with $q = h^3/12\mu \partial p/\partial \xi$). In these simulations it was observed that the higher water flow rate resulted in faster crevasse propagation, indicating that the choice of flow model can directly impact the time-scale over which the propagation process occurs. As altering the friction flow parameters can similarly alter the time-scale imposed by the water flow, the presented simulation will use a single set of friction flow parameters taken from Tsai and Rice (2010). This ensures that there is no artificial fitting of our results to observations by tuning the flow parameters, instead our approach relies on including physically relevant processes to better capture reality. Even though the values used for these parameters are realistic, we do acknowledge that better fits of observed results could be attained by fine-tuning these friction parameters.

The total aperture of the fracture is obtained based on the displacement jump across the discontinuity and melting layer thickness as:

$$h = h_\text{melt} + \mathbf{n} \cdot [\![\mathbf{u}]\!] \tag{8}$$

and the friction factor $f$ related to the (constant) wall roughness $k_\text{wall}$ and the reference friction factor $f_0$ as (Strickler, 1981):

$$f = f_0 \left( \frac{k_\text{wall}}{h} \right)^{\frac{1}{3}} \tag{9}$$

Together, Eqs. (7) and (9) allow for the fluid transport within the crack to be described explicitly as:

$$q = -2\rho_\text{w}^{-\frac{1}{2}} k_\text{wall}^{-\frac{1}{6}} f_0^{-\frac{1}{2}} h^{\frac{5}{3}} \left| \frac{\partial p}{\partial \xi} - \rho_\text{w} \mathbf{g} \cdot \mathbf{s} \right|^{-\frac{1}{2}} \left( \frac{\partial p}{\partial \xi} - \rho_\text{w} \mathbf{g} \cdot \mathbf{s} \right) \tag{10}$$

To conserve the meltwater within the fracture, this flux must satisfy the mass balance equation (Réthoré et al., 2006; Carrier and Granet, 2012; de Borst, 2017; Hageman and de Borst, 2021):

$$\frac{\partial q}{\partial \xi} + \dot{h} - \frac{\rho_\text{i}}{\rho_\text{w}} \dot{h}_\text{melt} + \frac{h}{K_\text{w}} \dot{p} = 0 \tag{11}$$

where the four terms account for the changes in fluid flux, additional volume created through deformations and melting, fluid produced by the melting process, and the fluid compressibility respectively. The compressibility term uses the bulk modulus of water $K_\text{w}$ to allow it to be slightly compressible. While this term is near negligible in practice (especially with the properties used in this paper), it provides a damping-like term within the numerical solution scheme and helps to stabilise the simulations. Through Eq. (11), the pressure within the crevasse is temporarily decreased when the crevasse opens, limiting the rate of crevasse opening and pressurisation based on the available fluid flow. This process introduces a timescale to the hydraulic fracturing process, which is often absent when a set melt water height is used (Clayton et al., 2022; Sun et al., 2021), wherein the pressure is directly prescribed based on the local water height relative to the crack tip.

To complete the model, a free inflow condition is imposed at the inlet of the fracture (i.e. at the crevasse mouth) through a penalty-like approach:

$$q_{\text{inlet}} = k_{\text{p}} \left( p_{\text{ext}} - p \right) \tag{12}$$

where $p_{\text{ext}}$ is the constant pressure at the inlet, taken equal to the hydro-static pressure of a 10-m deep lake, and $k_{\text{p}}$ is a
penalty parameter chosen large enough to accurately enforce this inflow constraint. We elect to enforce this pressure boundary condition in a weak sense through a penalty approach, such that the fluid influx at the inlet can be easily recorded. It should be noted, however, that using Eq. (12) with a high value for $k_{\text{p}}$ is equivalent to directly substituting in this pressure as boundary condition, and it has been verified that the pressure at the inlet is approximately equal to the applied $p_{\text{ext}}$. For post-processing purposes, this flux is additionally integrated over time to obtain the total volume of water flowing into the crevasse as:

$$Q_{\text{total}} = \int_{t} q_{\text{inlet}} \, \mathrm{d}t \tag{13}$$

As a two-dimensional domain is considered, this produces the total volume of fluid that has entered the crevasse per unit out-of-plane width, given in $\mathrm{m}^3/\mathrm{m}$. For the section where results are compared to observations from literature, this inflow per unit width is converted to total inflow by assuming a prescribed out-of-plane width of the crevasse.

Eq. (11) is solved to capture the macro-scale behaviour of the fluid contained within the fracture, whereas Eq. (10) defining the fluid flux together with the thermal effects described in the next section are solved to capture the micro-scale behaviour on a point-by-point basis. This novel and efficient two-scale approach is needed to account for the dynamic feedback between fluid flux and wall melting, which precludes a closed-form expression for the fluid flux.

### 2.2.2 Thermal model

The thermal processes within the fracture are described by assuming the heat transfer through advection within the crack to be negligible due to the near-freezing temperature of the lake water and limited opening width. As a result, the water within the fracture is required to be in thermal equilibrium with the surrounding ice, with any imbalance resulting in either additional ice melting or water freezing within the crack. This melting rate is described by:

$$\rho_{\text{i}} \mathcal{L} \dot{h}_{\text{melt}} - j_{\text{ice}} - j_{\text{flow}} = 0, \tag{14}$$

using the latent heat of fusion $\mathcal{L}$. Two contributions to the heat flux are considered in the above equation: (1) $j_{\text{ice}}$ represents the heat being released by the ice into the water, which is always negative due to the surrounding ice being below the freezing point by definition; (2) $j_{\text{flow}}$ is the heat flux produced through turbulent flow, which is always positive. If the turbulent heat flux is larger than the ice absorption, $j_{\text{flow}} > -j_{\text{ice}}$, the ice will melt, as will be the case for a warm ice sheet that is at a temperature close to ice melting point with a large fluid flux flowing through the crevasse. The opposite happens for a relatively cold ice sheet with little fluid flow, $j_{\text{flow}} < -j_{\text{ice}}$, where the fracture walls will start freezing. We emphasise that the thermal process is self-reinforcing: once the walls start melting a larger fluid flux is enabled, resulting in a larger amount of heat production, which leads to further melting of the walls. Conversely, when the walls start to freeze, the fracture will be more restrictive to fluid flow, causing the heat production to be more limited.

The flow produced heat flux is given by (Andrews et al., 2022):

$$j_{\text{flow}} = -q \left( \frac{\partial p}{\partial \xi} - \rho_{\text{w}} \mathbf{g} \cdot \mathbf{s} \right) \tag{15}$$

which scales with $(\frac{\partial p}{\partial \xi} - \rho_{\text{w}} \mathbf{g} \cdot \mathbf{s})^{3/2}$ and $h^{5/3}$, considering the definition of fluid flux $q$ in Eq. (10). As such, it is expected that this heat flux becomes significant for large opening heights, which are more easily achieved for thicker ice sheets due to the increased over-pressure sustained by the lake water. Furthermore, the gravity contribution to the driving force will cause increased melting for vertical cracks. Horizontal parts of the crack, in contrast, will exhibit reduced frictional heating, especially near the crack tips where the opening height and fluid flux are limited.

For the heat absorbed by the surrounding ice, it is assumed that once the fracture surfaces are created, heat conducts away normal to the fracture, described through the one-dimensional heat conduction equation:

$$\rho_{\text{i}} c_{\text{p}} \dot{T} - k \frac{\partial^2 T}{\partial \eta^2} = 0 \tag{16}$$

using the heat capacity $c_{\text{p}}$ and the thermal conductivity $k$, and using the surface-normal coordinate $\eta$. Because the ice-water boundary is consistently at $T_{\text{w}} = 0\,^{\circ}\text{C}$, an analytic solution for this equation is given by (White, 2006):

$$\frac{T(\eta,t) - T_{\infty}}{T_{\text{w}} - T_{\infty}} = \text{erfc} \left( \frac{\eta}{2\sqrt{\frac{k}{\rho_{\text{i}} c_{\text{p}}} (t - t_0)}} \right) \tag{17}$$

using erfc to indicate the complementary error function, $T_{\infty}$ the temperature of the ice, and $t_0$ the time from which heat conduction occurs, equal to the time of fracture. By taking the derivative with respect to $\eta$, and the temperature in $^{\circ}\text{C}$ (such that $T_{\text{w}} = 0\,^{\circ}\text{C}$), the thermal flux at any point with distance $\eta$ to the wall is given as:

$$j(\eta) = -k \frac{\partial T}{\partial \eta} = k \cdot \frac{2 T_{\infty}}{2\sqrt{k/\rho_{\text{i}} c_{\text{p}}} \sqrt{t - t_{\infty}}} \exp \left( -\frac{\eta}{2\sqrt{k/\rho_{\text{i}} c_{\text{p}}} \sqrt{t - t_{\infty}}} \right)^2 \tag{18}$$

where the factor $\pi$ within this equation is a result of the derivative of complimentary error function erfc from Eq. (17) and is not related to any axisymmetry assumptions, $\partial \text{erfc}(x)/\partial x = -2\exp(x^2)/\sqrt{\pi}$. Taking the gradient at the wall, $\eta = 0$, and accounting for the fact that each fracture contains two surfaces from which heat is absorbed, then results in the heat absorbed into the ice (White, 2006):

$$j_{\text{ice}} = -2k \frac{\partial T}{\partial \eta} = \frac{2 k^{\frac{1}{2}} T_{\infty} \rho_{\text{i}}^{\frac{1}{2}} c_{\text{p}}^{\frac{1}{2}}}{\pi^{\frac{1}{2}} (t - t_0)^{\frac{1}{2}}} \tag{19}$$

As a result, the temperature changes throughout the ice sheet do not need to be resolved, and it is instead sufficient to keep track of the time since the local fracture was created. This time-since-fracture is then used to determine the heat flux at the wall during the simulation, greatly reducing the computational cost, and can be used to estimate the temperature close to the fracture. It should be noted, however, that by assuming the heat conduction to be localised near and normal to the crack, the fracture always needs to propagate through ice at its initial temperature. As a result, once the fracture propagates, the new crack

surfaces start to release heat into the surrounding ice and the surfaces start to freeze instantly. This is in contrast to assuming heat to be conducted in all directions away from the crack, in which case the ice ahead of the crack tip will be partially heated, depending on the crack propagation speed. As such, the model described here is solely accurate if crack propagation exceeds heat conduction, which considering the low thermal conductivity of ice and small temperature differences is a fairly reasonable criterion.

By using analytic expressions for one-dimensional heat conduction into the ice instead of separately simulating the temperature throughout the ice sheet, we explicitly separate the thermal problem as a "micro-scale" from the mechanical problem, considered the "macro-scale". This assumes the thermal effects are localised near the crevasse, such that the overall mechanical material parameters (e.g. tensile strength and creep coefficient) are not altered by local changes in temperature, and that the overall geometry is not changed by the melting process in the fractures. To establish the validity of this separation of length scales between heat conduction and mechanical fracture propagation, the length-scale influenced by the heating due to the crevasse is estimated as $\ell_{\text{thermal}} = \sqrt{tk/\rho_i c_p}$, which for the two hour duration of our simulations is $\ell_{\text{thermal}} \approx 8$ cm. As this length scale is orders of magnitude smaller compared to the crevasse length (and well below the element length used), using this separation of scales is a good approximation. This furthermore shows the strength of the described two-scale model: if we were to explicitly simulate the thermal processes throughout the complete ice sheet and wanted to include the crevasse, our spatial discretisation would need to be fine enough to capture this thermal length scale, and would thus require centimetre-sized elements around the crevasse. Instead, by formulating the conduction through the analytic expression from Eq. (19) the thermal energy entering the ice is captured by the subgrid-scale formulation, and no separate discretisation is needed to accurately capture it.

Comparing the two heat fluxes, Eqs. (15) and (19), provides an indication whether the thermal processes are dominated by freezing or melting:

$$
\frac{j_{\text{flow}}}{-j_{\text{ice}}} = \sqrt{\frac{\pi}{k\rho_i c_p \rho_w f_0 k_{\text{wall}}^{1/3}}} \frac{(t-t_0)^{1/2} h^{5/3} \left| \frac{\partial p}{\partial \xi} - \rho_w \mathbf{g} \cdot \mathbf{s} \right|^{3/2}}{T_\infty}
\tag{20}
$$

with the first term being composed of physical constants, and the second term by case-dependent variables. The melting process dominates when $j_{\text{flow}}/-j_{\text{ice}} > 1$, coinciding with the case of large opening heights or warm ice sheets. The relevance of the melting process will also increase over time, as the surrounding ice warms up due to the presence of the water-filled crevasse. In contrast, for short timescales the freezing process will be dominant for almost all glacial temperatures, which will impose a limit on the ability of the hydraulic fracture to develop.

### 2.3 Implementation

The ice sheet fracture problem is described by the momentum balance in the domain $\Omega$, Eq. (1), and the mass balance within the fracture $\Gamma_d$, Eq. (11). These two equations are discretised using the finite element method, using quadratic quadrilateral elements for the displacements $\mathbf{u}$, and using quadratic interface elements for the fluid pressure $p$. Near the expected fracture path, these elements have a characteristic size of $2.5$ m, whereas away from the interface elements up to $20$ m are used. The temporal discretisation is performed using a Newmark scheme for the solid acceleration, and an implicit backward Euler

| **Ice:** | | | |
|---|---|---|---|
| Temperature | $T$ | Fig. 5 | |
| Young's Modulus | $E$ | 9 | GPa |
| Poisson Ratio | $\nu$ | 0.33 | |
| Density | $\rho$ | 910 | kg/m$^3$ |
| Creep coefficient | $A$ | Eq. (21) | |
| Reference creep coefficient | $A_0$ | $5 \cdot 10^{-24}$ | 1/Pa$^3$s |
| Creep activation energy | $Q_\mathrm{c}$ | 150 | kJ/mol |
| Creep exponent | $n$ | 3 | |
| Reference temperature | $T_\mathrm{ref}$ | 273.15 | K |
| Latent heat | $\mathcal{L}$ | 335000 | J/kg |
| Conductivity | $k$ | 2 | J/m s$^\circ$C |
| Thermal capacity | $c_\mathrm{p}$ | 2115 | J/kg$^\circ$C |
| Tensile Strength | $f_\mathrm{t}$ | Eq. (22) | |
| Reference strength | $f_\mathrm{t0}$ | 2 | MPa |
| Strength degradation | $f_\mathrm{deg}$ | $6.8 \cdot 10^{-2}$ | MPa/K |
| Fracture Energy | $G_\mathrm{c}$ | 10 | J/m$^2$ |

| **Rock:** | | | |
|---|---|---|---|
| Temperature | $T$ | 0 | $^\circ$C |
| Young's Modulus | $E$ | 20 | GPa |
| Poisson Ratio | $\nu$ | 0.25 | |
| Density | $\rho$ | 2500 | kg/m$^3$ |
| Creep coefficient | $A$ | 0 | 1/Pa$^3$s |
| Conductivity | $k$ | 2 | J/m s$^\circ$C |
| Thermal capacity | $c_\mathrm{p}$ | 770 | J/kg$^\circ$C |

| **Water:** | | | |
|---|---|---|---|
| Bulk Modulus | $K_\mathrm{w}$ | 1 | GPa |
| Density | $\rho$ | 1000 | kg/m$^3$ |
| Wall Roughness | $k_\mathrm{wall}$ | $10^{-2}$ | m |
| Reference friction factor | $f_0$ | 0.143 | |
| Surface pressure | $p_\mathrm{ext}$ | 0.1 | MPa |
| Inflow penalty factor | $k_\mathrm{p}$ | $10^6$ | m$^3$/s Pa |

**Table 1.** Material properties used within the simulations

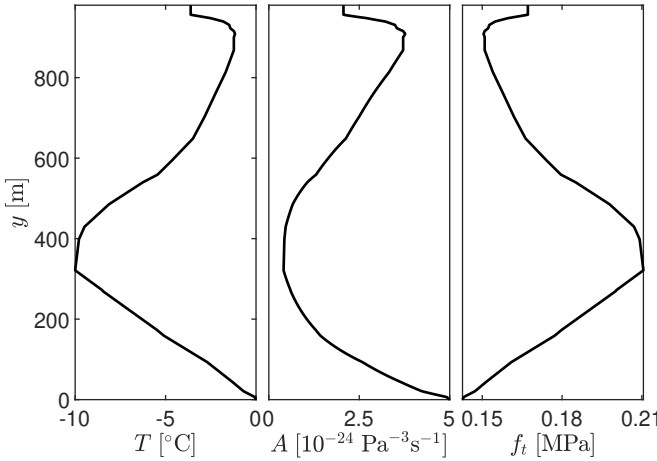

**Figure 5.** Depth dependence of the ice sheet temperature, and resulting creep coefficient and tensile strength.

scheme for the pressure, wall melting, and integration of quantities for post-processing (total water inflow, thermal fluxes). The only exception quantity not using an implicit time discretisation scheme is the viscous strains, which change slowly compared to all other variables within the system. While it is possible to include the strain increments in a time-implicit manner, we elect here to evaluate this term using an explicit Euler scheme and thus only require updating this quantity once at the beginning of each time increment. The resulting systems of Eqs. (8), (10) and (14), including implementation details for the "micro-scale" problem at the scale of the fracture opening , are provided in the supporting information.

The "macro-scale" governing equations at the glacier length scale (Eqs. (1) and (11)) are solved in a monolithic manner. This monolithic solver uses an energy-based convergence criterion $[\mathbf{f}_\mathrm{u} \; \mathbf{f}_\mathrm{p}][\mathrm{d}\mathbf{u} \; \mathrm{d}\mathbf{p}]^T < \epsilon$, placing equal importance on the convergence of the nonlinear CZM and the fluid pressure within the crack. Once the solution is converged, the stresses ahead of the crack along a pre-determined path (dashed line in Fig. 2) are calculated. The stresses are compared to the tensile strength to determine fracture propagation. If the crack propagates, a single new interface element is inserted and more iterations of the monolithic solver are performed (including checking for additional fracture propagation) to obtain a solution, where both unknown variables (i.e., displacement and pressure) are solved at the end of each time increment using the updated crevasse length. It is noteworthy that the path along which new interface elements are inserted is pre-determined, allowing the crevasse to only propagate straight down, and then splitting into two basal cracks that can only propagate sideways in a straight line. While the pre-determination of crack path and insertion of cohesive interface elements only between continuum finite elements are limitations of our numerical approach, it is reasonably realistic given the 2D idealisation of the rectangular glacier domain. The requirement of the crack path to be known *a priori* restricts the nucleation of crevasses elsewhere in the domain. Also, we do not model the surface hydrology associated with the formation of supra-glacial lakes, but rather assume a pre-existing lake with known depth that intersects with this initial crack.

To initialise the simulations, $1\,\mathrm{day}$ of time is simulated using time increments of $\Delta t = 10\,\mathrm{minutes}$, where the crevasse is not allowed to propagate. During this period, the viscous creep alters the stress state from that of a compressible linear elastic

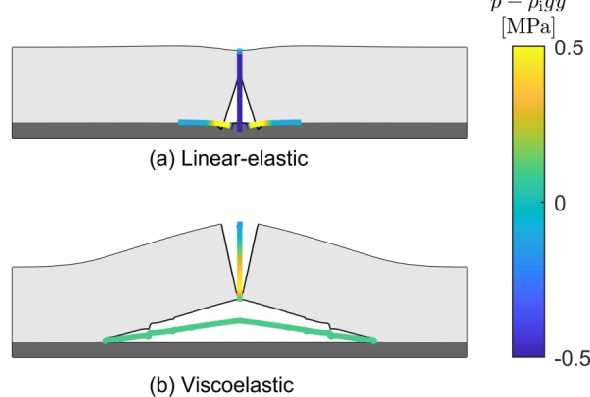

**Figure 6.** Over-pressure of the water within the crevasse relative to the cryostatic ice pressure, and deformations after 2 hours. Deformations in (a) and (b) are magnified by 1000 times. Animations showing the deformations over the full simulation duration are provided in the Supporting Information.

material to that of a nearly incompressible material compatible with viscoelastic deformations. This 1 day initialisation period was long enough for the stresses within the ice to stabilise to the steady-state, with further initialisation time not altering these stresses. After this initial time period, the crack is allowed to propagate and the remainder of the simulation is performed using $\Delta t = 2\,\mathrm{s}$ for a duration of $2\,\mathrm{hours}$.

### 2.4 Material properties

The properties used to model the ice and rock layers are provided in Table 1, along with parameters for water flowing in the crack. The temperature of the ice sheet is approximated using a temperature profile from the Jakobshavn glacier (Ryser et al., 2014b, a), with the location of this profile $\approx 80\,\mathrm{km}$ away from the lake drainage observations used as comparison. This temperature profile, shown in Fig. 5, is directly used within the description of the melting and freezing process. Additionally, the creep coefficient of the Glen's law, Eq. (3) is determined based on this temperature profile as (Weertman, 1983; Duddu and Waisman, 2012; Greve et al., 2014):

$$A = A_0 \exp\left(\frac{-Q_\mathrm{c}}{R}\left(\frac{1}{T} - \frac{1}{T_\mathrm{ref}}\right)\right), \tag{21}$$

and the tensile strength of the ice as (Litwin et al., 2012):

$$f_t = f_{t0} - f_\mathrm{deg}T. \tag{22}$$

These equations use the ice temperature $T$ and the reference temperature $T_\mathrm{ref}$ in Kelvin. They further use the activation energy related to the creep within ice, $Q_\mathrm{c}$, the gas constant $R$, and the temperature degradation rate $f_\mathrm{deg}$. The creep coefficient and tensile strength from these relations are shown in Fig. 5 for the used temperature profile.

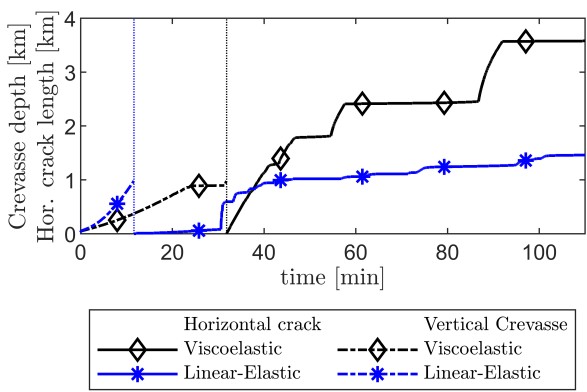

**Figure 7.** Vertical crevasse depth (dashed lines) and horizontal crack length at the ice-bedrock interface (solid lines) following the definitions from Fig. 1b. Vertical dotted line indicates the moment the vertical crevasse reaches the base of the ice sheet, after which point the horizontal crack propagation begins.

## 3 Results

### 3.1 Importance of viscoelasticity

The deformations of the ice sheet using a viscoelastic and linear-elastic constitutive model are shown in Fig. 6. If the linear-elastic model is used, crack opening is determined by the balance between the elastic stresses and the pressure within the crevasse Fig. 6a. This results in limited opening, and the vertical crevasse propagation does not consume a significant volume of lake water. The crevasse propagation rate is governed by this water flow, retaining a pressurised crevasse and reaching the bedrock layer after 13 minutes. Upon reaching the rock layer, once the pressure within the crevasse is significantly larger to start the horizontal crack propagation, the ice sheet will begin to lift up solely due to the water pressure acting at the base of the vertical crevasse. As ice is assumed to be compressible elastic, the higher water pressure causes the crevasse to be wider at its base and more narrow towards the ice surface owing to the lower water pressure. However, once the ice sheet starts to lift up significantly, the water pressure in the horizontal crack underneath the ice acts in the vertical direction opposing vertical cryostatic pressure. At this moment, the water pressure required to further propagate the horizontal crack decreases substantially. This causes a significant "burst" in the propagation, as shown in Fig. 7, but the additional volume created with the horizontal crack leads to the reduction in water pressure at the inlet. As the ice sheet is lifted up, the deformation of the ice sheet resembles that of a floating ice beam with the depth-varying water pressure applying a triangular distributed load on the vertical crevasse walls. The resultant bending moment causes the opening width at the upper portions of the crevasse to be restricted, and eventually force it to be fully closed, as illustrated by the blue solid line in Fig. 8. This prevents any further fluid from entering the crevasse, causing the only slight increases in pressure due to the freezing of fracture walls, and thus eventually stopping the propagation of the horizontal crack at the ice-bedrock interface.

In contrast, using the viscoelastic ice sheet rheology produces a crevasse that continuously widens during downwards propagation due to creep deformation induced by the overpressure within the crevasse. This means that part of the water entering

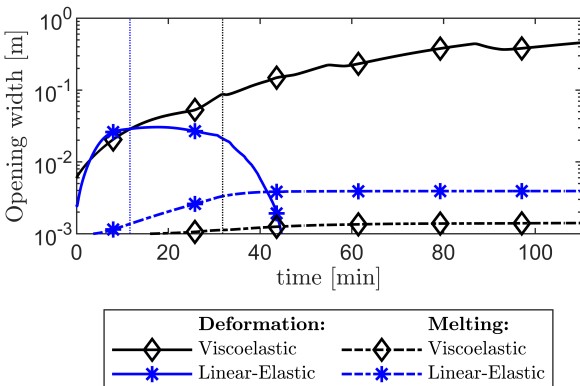

**Figure 8.** Crevasse mouth opening width due to viscous and elastic deformations in the surrounding ice and melting of the crevasse walls. Vertical dotted lines indicate the time when the vertical crevasse reaches the base of the ice sheet. Notably, the opening width is continues to increase in the viscoelastic case, whereas it decreases to zero in the linear-elastic case. Crevasse opening due to melting is at least an order of magnitude less than that due to deformation.

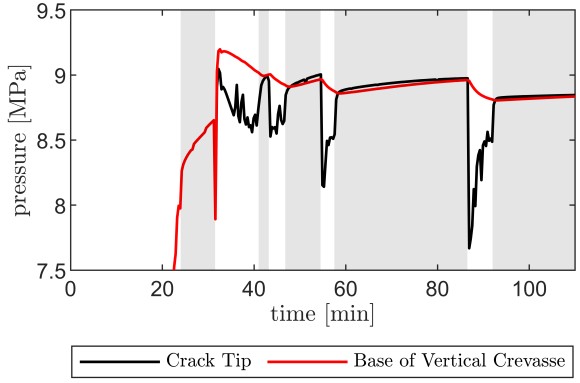

**Figure 9.** Pressure at the crack tip (black) and at the base of the crevasse (red) for the viscoelastic case. Shaded regions correspond to moments when the crack propagation is halted, as visible in Fig. 7.

the crevasse will need to be retained to compensate for the ever-increasing crevasse volume. As a result, the rate of downwards crevasse propagation rate is slower compared to the linear elastic model, which does not account for this additional opening

width. The viscoelastic creep strains also cause the opening of the crevasse to be ever-increasing while the crevasse is pressurised,Fig. 8, which allows more water to flow into the crevasse and downwards towards the bed. Once a crevasse penetrates the entire ice thickness and the crack tip reaches the bed, horizontal crack propagation commences, facilitated by the higher stresses due to the incompressible nonlinear viscoelastic response. Furthermore, as the ice sheet begins to uplift, even though the pressure within the crevasse suddenly drops due to the newly created volume, the viscoelastic creep deformations that

occurred during the downward propagation allow the vertical crevasse to remain open (note that the crevasse closes at this stage in the linear elastic model). As the crevasse remains open, water is able to flow through it, reach the bed, and uplift the ice sheet. The sustained transfer of water from the supraglacial lake to the bed is the main control on the rate of basal uplift and horizontal basal crack propagation. As the horizontal basal crack continues to propagate, the rate at which additional crack volume is created due to viscous deformations continues to increase while the rate of water inflow through the vertical

crevasse increases relatively slowly. This causes the pressure at the base of the vertical crevasse to decrease, as shown in Fig. 9. Eventually, this pressure becomes sufficiently low such that further crack propagation is paused, coinciding with a stress state where the viscous deformations allow for a crack volume enhancement equal to the water inflow. As the opening height of the vertical crevasse continues to increase due to viscous deformations, even when the horizontal crack is halted, the rate of water inflow slowly starts to exceed the rate of volume increase, allowing the pressure within the horizontal crack to slowly recover.

Once this pressure is sufficiently high, the horizontal crack resumes propagation. This alternation in pressure at the crack tip leads to episodic propagation. This process causes the undulations observed in the shape of the basal surface of the ice sheet (see Fig. 6) that are at most only $\sim 10\%$ of the total opening height, and the stepped crack length plot in Fig. 7 due to episodic propagation and halting.

## 3.2 Importance of melting

The thermal energy produced by friction, gained due to freezing, and lost due to conduction into the ice are given in Fig. 10. As the bedrock layer is at the melting point of ice (see Fig. 5), the only thermal energy loss due to conduction is through the vertical crevasse walls, whereas in the horizontal crack, no temperature difference exists to conduct heat into the ice. The conductive heat fluxes continue to increase proportionally with crevasse depth while the downward propagation is occurring, until the crevasse reaches the bedrock. After this time, thermal conduction is solely governed by the $t^{-3/2}$ dependency, causing

the conduction rate to slowly decrease as the ice surrounding the crevasse warms. As this is unrelated to the rheological model used for the ice, the conduction of thermal energy follows the same trend for both models, with the conduction energy loss for the viscoelastic model lagging slightly behind due to the crevasse reaching the base at a slower rate. In contrast, frictional heating shows a strong dependence on the rheology model. In the linear elastic model case, the smaller crevasse opening combined with the faster propagation causes significantly more frictional heating due to fluid flow at the initial stages.

However, as the fluid flow stops at later stages due to the closing of the crevasse mouth, this frictional heating also arrests. In the viscoelastic model case, the fluid has a wider crevasse to flow through, but only a limited amount of fluid flows into

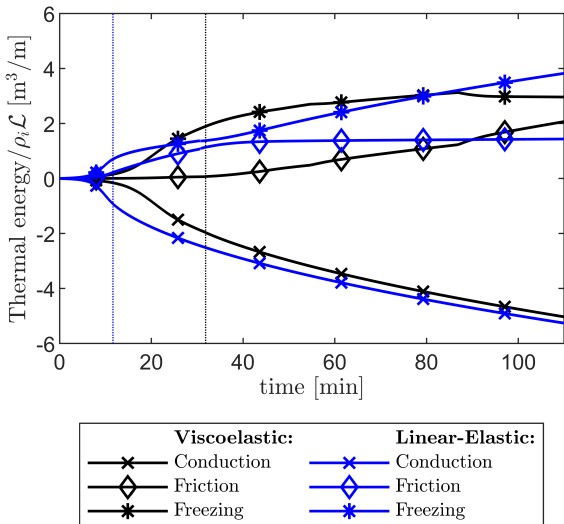

**Figure 10.** Total thermal energy created due to water-ice friction, lost due to conduction into the ice, and consumed by freezing (positive) or melting (negative). Energies are normalised by the heat $\rho_i \mathcal{L}$ required to melt $1\,\mathrm{m}^3$ of ice, and the values are calculated per unit out-of-plane length.

the horizontal crack at the glacier bed, which initially reduces the total thermal energy generated by friction (i.e. for $t < 90$ minutes in Fig. 10). However, because the fluid flow is continuously maintained due to wider crevasse opening, the length over which the fluid is transported increases as the horizontal crack propagates; therefore, the rate of heating increases throughout

the simulation, eventually exceeding the heating produced by the linear elastic model (i.e. for $t > 90$ minutes in Fig. 10).

The difference between the conduction and frictional heat fluxes dictates the rate of freezing at the crevasse walls. In the linear elastic model case, when the fluid flow stops the friction heat flux goes to zero, so freezing will occur at a steady rate at the vertical crevasse walls. Eventually, at a certain point in time, the vertical crevasse and the horizontal crack will fully refreeze, although this does not happen within the simulated time of 110 minutes shown in Fig. 10. In contrast, in the viscoelastic model

case, the ever-increasing rate of frictional heating and the slowly reducing rate of conductive losses cause the freezing process to slow down, and eventually cause some of the initially frozen crevasse walls to start melting. It should be noted, however, that the thermal energies produced due to friction and lost due to conduction are quite small, only sufficient to freeze $\approx 2\,\mathrm{m}^3/\mathrm{m}$ of ice over the full length of the crevasse ($980\,\mathrm{m}$ downwards, and up to $4\,\mathrm{km}$ sideways). This amounts to 1–2 mm of ice building up on the vertical crevasse walls over this time period, compared to a crevasse opening width of $\approx 0.5\,\mathrm{m}$ as shown in Fig. 8.

This differs from the crevasses observed in reality, where opening widths of several meters are not uncommon. While melting is commonly credited with partly creating such large openings (Andrews et al., 2022), our results allude to processes such as rapid glacial sliding (due to reduced basal friction) occurring after the onset of the lake drainage event could lead to large crevasse openings.

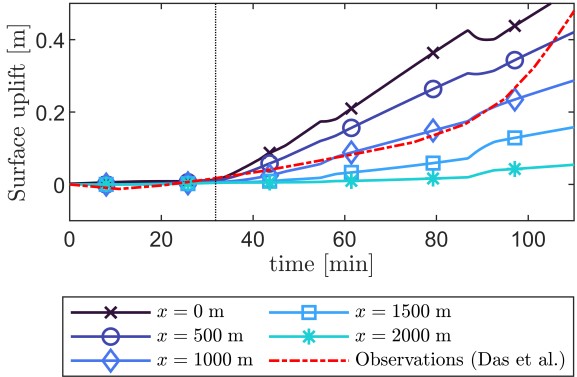

**Figure 11.** Vertical uplift of the ice sheet surface over time at set distances from the crevasse ($x = 0$ m is at the crevasse and $x = 500$ m is 500 meters to the left/right of it), using the viscoelastic rheology. For comparison, observational data from Das et al. (2008) is included. The vertical dotted line indicates the moment the crack reaches the base of the ice sheet. (Linear-elastic uplift provided in the supporting information)

### 3.3 Application to rapid lake drainage

We conduct a comparative study with the observations from Das et al. (2008), as shown in Fig. 11. To obtain these changes in the water level from the two-dimensional simulation results (producing fracture inflows in $\mathrm{m}^3/\mathrm{m}$), we assumed the estimates of the lake area as $A_{\mathrm{lake}} = 5.6 \, \mathrm{km}^2$ and the out-of-plane length of the crevasse as $W_{\mathrm{oop}} = 3.2 \, \mathrm{km}$ (based on values estimated by Das et al. (2008) and used by Tsai and Rice (2012)), allowing the water level change to be obtained as $\Delta h_{\mathrm{lake}} = Q W_{\mathrm{oop}} / A_{\mathrm{lake}}$. As our model predicts horizontal cracks of 1.6 km in each direction, compared to the out-of-plane width of $W_{\mathrm{oop}} = 3.2 \, \mathrm{km}$,

assuming plane-strain conditions is reasonable for this case. We calculated fluid inflow rates using the lake height time series in Das *et al.*, which has a 20 minute sampling frequency. Due to this sample frequency, the water level changes consist of a linear interpolation between sample points, while the inflow rate is only piece-wise continuous.

Even though the 2D model considered here is highly idealised and does not represent the complex, 3D crevasse and lake geometry in reality, the predicted surface uplift matches well with the measurements of a GPS station located 1–2 km from

the crevasse for the first $90 \, \mathrm{minutes}$ (Fig. 11). In contrast, the observed and simulated lake water-level changes match the observations to a lesser extent, as shown in Fig. 12. One potential explanation for the larger mismatch in the fracture inflow compared to uplift is the observed development of secondary cracks during the hydraulic fracturing process, which could have significantly enhanced fluid transport to the ice sheet base. Additional assumptions for the numerical model are that the ice sheet is pristine (i.e. undamaged) and the ice-rock boundary is initially frozen, neither of which is strictly correct. Within the ice,

pre-existing cracks, crevasses, and defects can link to the newly developed hydrofractures, which could significantly enhance the water inflow. Furthermore, fluid flow and movement at the ice-bed interface as it drains downglacier could influence the modelled fluid inflow and ice sheet uplift. These connections are also a potential explanation for the exponential increase in lake drainage, which differs from the behaviour observed within the simulations. One final point of potential mismatch is the

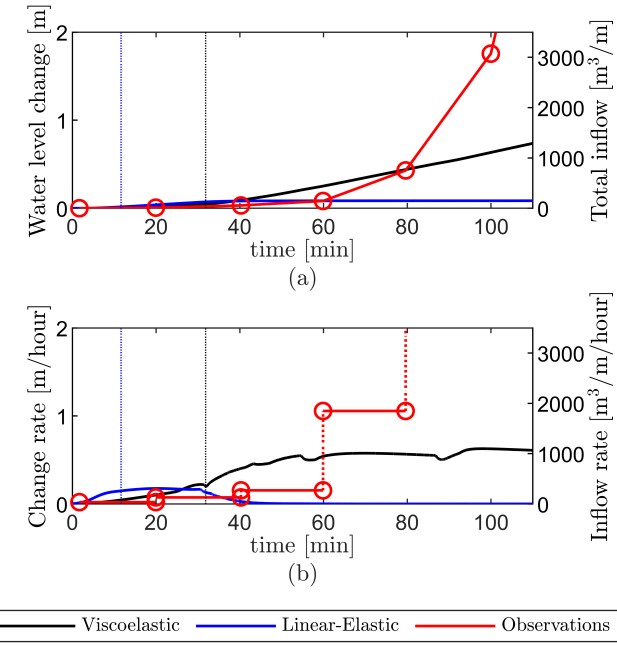

**Figure 12.** Model results using a linear-elastic (blue) and viscoelastic (black) ice rheology are compared to the reference observations (red, circles) from Das et al. (2008): (a) lake water level versus time (b) rate of water level change versus time. The vertical dotted line indicates the moment the vertical crevasse reaches the base of the ice sheet. For all lines, the vertical axis report the values in $\mathrm{m}^3/\mathrm{m}$, as directly resulting from simulations, on the right and on the left in m water-level change, converted using $A_{\mathrm{lake}} = 5.6\ \mathrm{km}^2$ and $W_{\mathrm{oop}} = 3.2\ \mathrm{km}$.

conversion between water volumes resulting from our simulations to the lake drainage height reported by Das et al. (2008)
(and inversely, from lake water level to volumes from Das *et al.*). We assume a simplified lake geometry with a constant area as water height decreases instead of taking into account lake bathymetry. This simplification is therefore likely responsible for the model's underestimation of lake water level change particularly during the later stages of the lake drainage.

## 4 Discussion

### 4.1 Influence of rheology

With the linear elastic rheology, the vertical crevasse is able to propagate through the $980\ \mathrm{m}$ thick ice sheet in $13\ \mathrm{minutes}$; whereas, with the elastic-viscoplastic rheology it takes about $25\ \mathrm{minutes}$. Despite this time difference, only a small amount of lake water ($< 100\ \mathrm{m}^3/\mathrm{m}$) is required to drive the fracture to the base. Upon reaching the base and lifting up the ice sheet, the fracture propagation process is able to consume a substantial amount of water ($> 1000\ \mathrm{m}^3/\mathrm{m/hour}$) from the supraglacial lake. In contrast, with the viscoelastic rheology horizontal fracture propagation is episodic, characterised by periods of fast crack
growth and stagnation. Specifically, horizontal fracturing stagnates when fluid inflow is fully accommodated by the space created due to viscoelastic creep deformation, limiting any increase in fluid pressure necessary to drive further propagation.

However, as creep deformation widens the vertical crevasse, the increased fluid inflow leads to pressure increase and further fracture propagation.

A key difference is that with the viscoelastic rheology model, the method can capture the uplift of the ice sheet due to supraglacial lake drainage, consistent with observations; whereas, with the linear elastic rheology, uplift stagnates approximately after 1 km. This is because, in the linear elastic case, the depth-varying water pressure within the vertical crevasse and the resultant bending moment are sufficient to close the narrow crevasse opening at the top surface, thus preventing additional water from entering the vertical crevasse (see Figs. 6, 8 and 12). Our parametric studies (see supporting Fig. SI1-SI2) show that the bending effect in the linear elastic case is independent of ice thickness over shorter timescales (hours) and will always close the crevasse near the ice surface thereby stopping crevasse propagation.

While we only considered a long crevasse (using plane-strain assumptions), it is expected that these conclusions regarding the role of viscous strains hold if axisymmetric models were used or if the crevasse geometry is more complex: Irrespective of the crack geometry, the over-pressure within the crevasse will always result in viscous strains enhancing the opening, increasing the water transport towards the glacier bed. High stress concentrations in the area surrounding the crack tip will also still result in a short Maxwell time-scale, such that viscous strains occur on time-scales comparable to those relevant to hydro-fracture. While the horizontal crack growth rate, surface elevation change (uplift) and water level change may be different, the main finding - viscous deformation exerts a much stronger control on hydrofracture propagation compared to thermal effects - would not change.

## 4.2  Influence of ice temperature

Thermal processes within the crevasse have a limited effect in our simulation studies (see Fig. 8). The change in crevasse opening due to freezing/melting (a few mm) is orders of magnitude smaller than the elastic/viscoplastic deformation (tens of cm). Consequently, frictional heating-induced melting is unable to prevent crevasse closure in the linear elastic simulations. On the flip side, conduction loss-driven freezing at the crevasse walls is not enough to close the crevasse opening in the viscoelastic simulations under the thermal conditions considered here. This is not to say that thermal effects are insignificant in the complete process: The creation of supraglacial lakes is driven by melting, and results show that if the surface layers of ice are cold enough crevasse propagation is halted before it reaches significant depths (see supporting information 3). We identify an ice temperature threshold of $-8°C$ at and below which crevasses freeze shut and propagation is prevented. Our results suggest that crevasse propagation models applied to rapid supraglacial lake drainages in warmer ice regions do not need to account for thermal interactions (causing refreezing) on short timescales of crevasse development. This finding can greatly simplify the parametrisation of coupled surficial and subglacial hydrology within models. For example, a simple parametrisation could involve identifying regions where ice surface temperatures are above -8°C and in these regions introduce the influence of surface melt on subglacial hydrology and basal friction.

### 4.3 Relating to observational data

Advances in technology have been reflected in the production of observational data sets capturing transient lake drainage events (Andrews et al., 2018; Chudley et al., 2019; Das et al., 2008; Doyle et al., 2013; Hoffman et al., 2011; Mejía et al., 2021; Stevens et al., 2015, 2022; Lai et al., 2021). However, models seeking to represent hydraulic fracturing still use analytical linear elastic fracture mechanics from the 1970s (Weertman, 1971, 1973) and 1990s (Desroches et al., 1994). The computational framework developed here advances the analysis of rapid lake drainage and fracture events. Our model results using a viscoelastic rheology produce realistic timescales for supraglacial lake drainages on the Greenland Ice Sheet and are closely tied to lake volume. Based on the total inflow at 120 mins (about 1300 m$^3$/m in Fig. 12a), we estimate that for the North Lake in 2006 (Das et al., 2008) $0.0042 \, \mathrm{km}^3$ of water drained in 2 hours through a crevasse with an out-of-plane length of $W_\mathrm{oop} = 3.2 \, \mathrm{km}$. Notably, the inflow rate of lake drainage (black line in Fig. 12b) stabilised to $0.0032$–$0.0035 \, \mathrm{km}^3/\mathrm{h}$ ($1000$–$1100 \, \mathrm{m}^3/\mathrm{m}/\mathrm{h}$) over the second hour of the simulation. Using this average inflow rate and assuming the same out-of-plane crevasse length reported in 2006, we now compare our results to observations of North Lake drainage from 2011–2013 (Stevens et al., 2015). Lake volumes and drainage duration over these three years were $0.0077 \, \mathrm{km}^3$ over $3 \, \mathrm{hours}$ in 2011, $0.0077 \, \mathrm{km}^3$ over $5 \, \mathrm{hours}$ in 2012, and $0.0057 \, \mathrm{km}^3$ over $5 \, \mathrm{hours}$ in 2013. Assuming that the inflow rate at 2 hours will remain constant for the next hour Fig. 12a, we obtain a total inflow volume of $0.0074 \, \mathrm{km}^3$ after 3 hours, which matches the reported values for 2011, albeit overestimating the lake volumes for the 5 hour drainage events from 2012 and 2013. The discrepancy in estimating lake volumes in 2012 and 2013 is likely related to the assumption of out-of-plane crevasse length, which leads to smaller inflow rates compared to 2011. Estimates of crevasse volume obtained from a network inversion filter for 2012 and 2013 are smaller than that for 2011 (Stevens et al., 2015).

We can also compare the rate of crevasse opening obtained within simulations to observed strain rates and slip velocities, as one of the assumptions in the computational model is that no basal slip occurs before basal uplifting occurs. Ice surface velocities that have been reported in the area surrounding the North Lake are around $91 \, \mathrm{m/year}$ (Ryser et al., 2014b), or around $1 \, \mathrm{cm/hour}$ (in the absence of lake ongoing lake drainage events). Das et al. (2008) reported speed-ups of this velocity by $300\%$ during lake drainage due to reductions in basal friction. However, even at this increased rate, the enhancement to the crevasse width would be limited to only 6 cm compared to the 50 cm crevasse opening created due to the water pressure within the crevasse inducing local elastic and viscous deformations. For basal motion to have a significant influence on crevasse width, lake water would need to reach the bed and become distributed over a large enough area to influence basal water pressures, uplift the ice, and cause enhanced sliding. Because of these prerequisites peak basal motion would have a delayed influence on crevasse width, only occurring after the crevasse has opened and transferred water to the bed. As this occurs well outside the considered time-span within this work (2 hours), the effect of basal motion on crevasse width can indeed be ignored in our study.

### 4.4 Future of Greenland ice sheet

As supraglacial lakes expand inland and occupy higher-elevation areas in response to atmospheric warming (Howat et al., 2013), understanding the influence of these lake drainages on subglacial hydrology is becoming more important. We foresee

this model's application to cases where describing the fracture mechanics in high detail is desired such as in constraining the formation and drainage of subglacial flood waves following rapid supraglacial lake drainages. While the presented model is restricted in scope to only solve for a single hydro-fracture and the resulting uplift, results can be coupled with data sets and other process based models to fully investigate ice dynamic response to rapid supraglacial lake drainages. For example, crevasse formation models (e.g., Hoffman et al., 2018) or remote sensing data can constrain the timing of supraglacial lake drainages with the later also defining pre-drainage lake conditions. Results from our subsequent model runs can be fed back into large-scale low-resolution glaciological models, or smaller floodwave propagation models (e.g., Lai et al., 2021) to inform changes in basal conditions both locally and downstream along the resulting subglacial floodwave. Indeed, when interpreting GPS-derived ice motion of supraglacial lake drainage events, utilising a high-resolution model such as this would enable one to parse out ice motion produced by fracture propagation and that produced by subglacial processes such as frictional sliding. Large volumes of water injected into the bed following rapid lake drainages create a subglacial floodwave that modulates ice velocity and can alter the subglacial drainage system as it moves downglacier. These floodwaves can connect and drain previously hydrologically isolated regions of the subglacial drainage system that control minimum ice velocities, thereby imparting lasting effects on hydrodynamic coupling evident throughout the remainder of the melt season (Mejía et al., 2021). The area of the ice sheet's bed that can be modified is unconstrained but can extend tens of kilometres downglacier. While our model does not incorporate a basal drainage component nor simulate the subglacial floodwave produced by the lake drainage event, these complex processes can be investigated in the future by coupling our model with a subglacial hydrology model to more fully assess the ice-dynamic and hydrological consequences of rapid lake drainages. The model presented here therefore not only provides a mathematical framework for interpreting in situ observations, but also provides a mechanism to simulate detailed subglacial flooding, which can provide more accuracy when inferring subglacial transmissivity and establishing initial conditions of the subglacial floodwave produced as water drains down glacier after the lake drainage event.

### 4.5 Advancing ice sheet modelling

Crevasses play a role in two of the most poorly understood glaciological processes, subglacial hydrology and iceberg calving; consequently, they are also poorly represented in ice sheet models. The impact of subglacial hydrology on basal motion (Bueler and van Pelt, 2015) remains poorly or rarely represented in numerical ice sheet models, leading to potentially large model uncertainty (Aschwanden et al., 2021). Data assimilation techniques typically use observations of GrIS geometry and assume ice rheology is known (based on ice temperature, englacial properties, crystal orientation and impurities), so that basal friction can be treated as the only unknown field parameter (Goelzer et al., 2017). While such techniques can provide modelled velocities close to observations in diagnostic simulations, nonphysical responses may be predicted in prognostic simulations (Seroussi et al., 2011). Existing ice sheet models (Lipscomb et al., 2019) do not incorporate changes in basal friction due to supraglacial lake drainage events, which can severely limit their applicability to future warmer scenarios. As we scale up our computational framework to glacier scale simulations, we intend to use it develop simpler parametrisations linking surface hydrology to subglacial hydrology. Future studies should consider mapping the combinations of strain rates and surface temperatures that relate to lake formation and drainage, and introduce fluid volume and pressure in fractures as inputs into

subglacial hydrology and basal friction models. Although this is not such a simple task, our two-scale modelling framework is a first step towards exploring the interactions between thermal, hydraulic and mechanical process controlling GrIS flow and fracture.

## 5   Conclusions

We have developed a two-scale computational method able to capture the hydraulic fracturing process responsible for rapid supraglacial lake drainages in high temporal and spatial resolution. By resolving ice sheet deformation in 2D, capturing the water within the crevasse in 1-D, and using a sub-grid-scale formulation based on analytic expressions for thermal conduction, melting, and frictional heating, the relevant mechanisms surrounding crevasses in ice sheets are all captured even though they are relevant on drastically different length scales. By separating these mechanisms across scales we achieve a highly (computationally) efficient scheme capable of capturing the dynamic processes of crevasse formation and subsequent ice uplifting. While no direct verification studies have been performed with this two-scale formulation for glacial fracturing, previous applications have demonstrated its accuracy for benchmark hydro-fracture cases in rock media, albeit without the thermal model (Hageman and de Borst, 2022, 2019).

Our novel modelling framework allows us to explore the vertical and horizontal fracture propagation during rapid lake drainage events, discern the role of thermal effects, and more importantly, evaluate the repercussions of ignoring viscoelasticity even on the short timescales associated with these events. We find that viscoelastic creep deformation has a significant effect on the horizontal fracture propagation at the glacier bed and the vertical crevasse opening width, allowing for a greater fluid flow required for rapid lake drainage. Our findings reasonably illustrate the time scales and the order of magnitude of water volumes involved in the rapid drainage of supraglacial lakes, as observed by Das et al. (2008). This study demonstrates the utility of our modelling framework for developing a parameterisation of hydraulic fracture in a large scale model of the Greenland ice sheet, and ultimately understand the implications for sea level rise.

*Acknowledgements.* T. Hageman acknowledges financial support through the research fellowship scheme of the Royal Commission for the Exhibition of 1851. E. Martínez-Pañeda acknowledges financial support from UKRI's Future Leaders Fellowship programme [grant MR/V024124/1]. R. Duddu acknowledges funding support from the NSF Office of Polar Programs via CAREER grant no. PLR-1847173, and The Royal Society via the International Exchanges programme grant no. IES/R1/211032. J. Mejia acknowledges support by the Heising-Simons Foundation #2020-1910. The authors also acknowledge computational resources and support provided by the Imperial College Research Computing Service (http://doi.org/10.14469/hpc/2232).

*Author contributions.* Tim Hageman: Software, Formal analysis, Data Curation, Investigation, Writing - Original Draft, Visualisation, Methodology. Jessica Mejia: Formal analysis, Investigation, Writing - Original Draft. Ravindra Duddu: Investigation, Writing - Original Draft, Writing - Review & Editing, Conceptualisation, Methodology. Emilio Martínez-Pañeda: Writing - Review & Editing, Conceptualisation.

*Competing interests.* The authors declare that they have no known competing financial interests or personal relationships that could have appeared to influence the work reported in this paper.

*Code availability.* The *MATLAB* code described in the methods and used to generate the results for this research is available from https:
630   //github.com/T-Hageman/MATLAB_IceHydroFrac, where documentation for this code and post-processing scripts are also provided. [code repository currently set to private, but will be made available at publication]

*Video supplement.* Animations showing the deformations and pressure over time for Fig. 6

**Supporting Information Appendices**

(1) Implementation details of the finite element scheme. (2) Parametric study regarding the effect of ice sheet thickness on the
closing of crevasses using linear elastic and viscoelastic rheologies. (3) Parametric study on the role of ice sheet temperatures on stagnating crevasse propagation due to freezing. (4) Figure showing the uplift over time using a linear elastic rheology (the linear elastic counterpart of Fig. 11). (5) Animations showing the deformations and pressure over time for Fig. 6.

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
