# Peer review of "Ice viscosity governs hydraulic fracture causing rapid drainage of supraglacial lakes"

_EGUsphere, 2024_

## Author Comment (AC1)

**Authors' response to reviewers report**

**Response to Reviewer 1**

This paper models the opening of hydraulic fractures below supraglacial lakes, and the subsequent spreading of the basal fracture, uplifting the overlying ice. The aim of the paper is to investigate the role that different rheological models for the ice have on the system. Results are compared with field data from Das et al. 2018 for a lake drainage event. The study finds that the inclusion of viscous creep in the ice is important to accurately capture something close to the observed behaviour: an elastic ice model does a bad job, even though the dynamics occur over a short timescale (a few hours) - presumably because of the large stresses involved.

This is an interesting study and I think the conclusions are useful for the community. However, there are quite a few points that I think could be explored in more detail, and the model results need a bit more exploration to be entirely convincing. I think it could use some fairly significant revisions, as outlined below, to increase its impact, and also to alleviate various issues of unclarity or inaccuracy.

We thank the reviewer for their detailed comments and assessment of the manuscript. Below we address the main concerns of the reviewer on a point-by-point basis, with changes highlighted in blue within the manuscript. However, we would like to directly address a reviewer comment raised in several points, regarding more and extensive simulations using different assumptions needed for the results of this modelling study to be convincing.

The key suggestions of the reviewer are to perform simulations with a purely viscous rheology (using a different set of governing equations than described in this manuscript to model ice as a fluid, instead of as a visco-elastic solid), considering axisymmetry (i.e., round conduits) instead of 2D planar fractures for the vertical crevasse, and considering different descriptions of the fluid flow within crevasses (ranging from laminar flow, to different approximations for turbulent flow). While these suggestions would indeed provide relevant and interesting results, they would not add to the main message of the article, that viscous deformations in addition to elastic deformations are important to consider within the context of rapid lake drainage due turbulent hydrofracture.

Currently, the revised article is 30 pages (+10 pages of supplementary materials), well exceeding the recommended length for The Cryosphere articles of 12 pages. Although changing the turbulent flow model (or tuning its parameters) potentially result in a better fit to observed events, investigating this would require an extensive description of the comparison between different flow models. It would, however, not add arguments to the finding that viscous deformations are relevant, even on short time-scales, but instead add a new set of findings that could be a research article on its own (potentially, that the fracture flow model provides the dominant time-scale in the crack propagation process). Similarly, to consider a purely viscous rheology and describe the results in a transparent, reproducible and understandable manner would require a second section for the governing equations, constitutive relations for ice as a non-Newtonian fluid, and extensive results and comparisons. As such, our view is that the extensions suggested by the reviewer would be better included in a separate stand-alone research paper. Nevertheless, we greatly appreciate the reviewer for their excellent insights and expertise on this topic.

Comment # 1

1) The problem is studied in two dimensions, and the authors go to some effort throughout to argue that this is a reasonable limit to consider. The other simple 'end-member' option would be a radial (i.e. axisymmetric) profile, as mentioned around line 75. I don't fully follow the reasoning in the paper here: there are statements that don't make sense, like "While our 2D model for the horizontal basal crack propagation and the basal uplift is valid for the axisymmetric assumption..." Presumably this should mean something like "The 2D planar model construction could be straightforwardly adapted to describe instead an axisymmetric spreading". The point is, the construction is not complicated, but there are different- and non-trivial - geometric factors which change some details.

Fundamentally, it seems to me that it should be straightforward to do the whole problem in an axisymmetric geometry as well as the 2D planar geometry - because it uses the same ideas and is still mathematically 2D. And this would be very valuable, because 'reality' is somewhere between the two (planar and axisymmetric; although arguably it is closer to axisymmetric than planar) and so comparing solutions for each would greatly help the impact of this work.

It is not straight-forward to conduct the full study as an axisymmetric geometry. For the sideways basal crack propagation and uplifting and the description of the viscoelastic ice, the proposed model can easily be adapted (as suggested by the reviewer) to axisymmetric cracks, instead of considering planar cracks; this will only require changes within the finite element implementation to consider radial symmetry (updating the definitions of the displacement-to-strain mapping operator, and integration weights to take into account the axisymmetry), while

the majority of the constitutive choices within the model remains unchanged. We have clarified when it would be appropriate to use axisymmetry in the paper by adding this sentence: *"using an axisymmetric representation would be appropriate for crevasses with surface lengths (i.e., the out-of-plane direction from Fig. 1b) much shorter compared to the length of the horizontal/radial basal crack, such that the vertical crevasse can be considered as a conduit propagating downwards. In contrast, the plane-strain representation used here is suitable for when the surface crevasse spans longer distances, such that the crack is considered as a plane propagating downwards."*

However, for the vertical crevasse the physical considerations are considerably different: Under a plane-strain assumption, the vertical crevasse is a crack with its propagation governed through the balance of elastic stresses and fracture toughness, dissipating energy due to the creation of fracture surfaces (with the amount of elastic energy dissipated scaling with $G_c A$, where $G_c$ is the fracture release energy and $A$ the total surface area of the crevasse). In contrast, if we assume axisymmetry, the downwards crevasse is now a cylindrical conduit, where it is no longer clear as to what fracture surface area is in the undeformed state and the corresponding the energy dissipation are. As these two processes are fundamentally different, no straightforward comparison of simulations can be preformed.

We have added the following clarification to the paper: *"This is an important distinction because of the associated processes governing fracture propagation. For a typical planar or "vertical" crevasse formed under plane strain, mechanical stresses drive crevasse propagation. In contrast, under axisymmetric conditions, the cylindrical conduit would propagate downwards due to fracture, melting and erosion processes. While it is possible to model it as a cylindrical moulin evolution (Trunz et al., 2022), it is difficult to describe the fracture mechanics using the existing framework. Of course a 3D model able to capture both these phenomena would be ideal, but it would be computationally too expensive, so we utilize the 2D plane strain approximation, focusing on the propagation of fractures driven by stresses and only consider the lateral melting of the fracture faces to ultimately align with the observed morphology of crevasses."*

Comment # 2

2) The results in figure 7 are concerning as it stands. The curves - particularly the black curve - are curiously non-smooth, and the mechanisms / reasons for this are not at all clear. There is some discussion around line 388 about this, but the explanation is not very convincing, and much more evidence needs to be presented to convince the reader this is not some funny numerical artefact. I can't see which aspect of the mathematical formulation is giving rise to this behaviour - for example, the black line goes flat for a period, and then increases suddenly. Is this robust to numerical resolution? What aspect of the model allows the crack to halt propagation for a period and then restart motion? What physics is controlling the length of time the crack is stationary for, and indeed, the time at which it decides to become stationary? Much more convincing analysis is needed of this behaviour. The point is discussed again around line 450, but again I don't see how the model is giving this 'episodic', almost stick-slip-like behaviour. Perhaps a plot of something like the pressure at the tip of the spreading crack would help to explain this phenomenon.

This step-wise propagation is a well-reported phenomena for fluid-pressure driven fracture [1-7]. As the crack propagates, the fluid inside of the crack is "redistributed" based on the new crack length, moving the fluid from the base of the vertical crevasse towards the horizontal crack tip. This causes the pressure surrounding the crack tips to change from negative (due to the newly created empty volume), to positive so that the crack can propagate again. As this redistribution is fairly fast, the crack continues to propagate smoothly as long as the pressure in the remainder of the crack does not get reduced too much. Re-pressuring the crack through the vertical crevasse is much slower (especially when new volume is created due to viscous deformation) causing the pauses in propagation.

This is shown in Fig. 1, showing the pressure oscillations at the crack tip and the pressure at the base of the vertical crevasse. The pressure at the crack tip (black line) shows oscillations due to the choice of element size in our discretisation, depending whether the crack propagated within that time increment or in earlier time increment, but this does not affect crack propagation results. In contrast, the pressure at the base of the vertical crevasse (red line) is fairly steady and does not directly show such oscillations. It also shows that as the crack propagates (the non-shaded regions) the pressure at the base decreases, indicating that the volume created by the propagation does not directly get filled by water from the surface, but rather with water stored elsewhere in the basal crack. Therefore, the pressure at crack tip (black line) during propagation is generally lower compared to the pressure at the base of the vertical crevasse (red line) due to the additional volume created during the crack opening needing to be water-filled. As the overall water pressure decreases, a pressure distribution is reached where the crack can no longer propagate, and the propagation halts (with a slight delay due to inertial effects).

Even when the crack is halted, the crack volume continues to increase due to viscous deformations. This causes the very slow re-pressurisation of the crack observed in Fig. 1, where the main increase in pressure is a result of increased water inflow rates due to the vertical crack width also increasing through viscous deformations. Eventually,

[Figure]

Figure 1: Pressure at the crack tip (black), and at the base of the horizontal crack (red) using a viscoelastic rheology. Shaded regions indicate that the crack propagation is paused at these moments.

this inflow increases sufficiently to increase the pressure throughout the basal crack, and propagation is resumed. it should be noted that this effect becomes stronger for longer cracks, as the uplift allowed due to viscous deformations enhances the crack volume more compared to for shorter cracks, hence the rate of pressure change (i.e., the slope of the red line) decreases at longer times in the simulations.

To convey this reason for the pauses in fracture propagation, we've added Fig. 1 and the above explanation (in a concise form) to the paper: *"As the horizontal basal crack continues to propagate, the rate at which additional crack volume is created due to viscous deformations continues to increase while the rate of water inflow through the vertical crevasse increases relatively slowly. This causes the pressure at the base of the vertical crevasse to decrease, as shown in Fig. 9. Eventually, this pressure becomes sufficiently low such that further crack propagation is paused, coinciding with a stress state where the viscous deformations allow for a crack volume enhancement equal to the water inflow. As the opening height of the vertical crevasse continues to increase due to viscous deformations, even when the horizontal crack is halted, the rate of water inflow slowly starts to exceed the rate of volume increase, allowing the pressure within the horizontal crack to slowly recover. Once this pressure is sufficiently high, the horizontal crack resumes propagation. This alternation in pressure at the crack tip leads to episodic propagation."*

[1] Schrefler, B. A., Secchi, S., Simoni, L.: On Adaptive Refinement Techniques in Multi-Field Problems Including Cohesive Fracture. Computer Methods in Applied Mechanics and Engineering (2006) https://doi.org/10.1016/j.cma.2004.10.014

[2] Pizzocolo, F. and Huyghe, J. M. and Ito, K.: Mode I Crack Propagation in Hydrogels Is Step Wise. Engineering Fracture Mechanics (2013) https://doi.org/10.1016/j.engfracmech.2012.10.018 [3] Secchi, S., Schrefler, B. A.: Hydraulic Fracturing and Its Peculiarities. Journal on Computational Engineering (2014). https://doi.org/10.1186/2196-1166-1-8

[4] Milanese, E and Rizzato, P and Pesavento, F and Secchi, S and Schreffler, B. A.: An Explanation for the Intermittent Crack Tip Advancement and Pressure Fluctuations in Hydraulic Fracturing. Hydraulic Fracturing Journal (2016). https://doi.org/

[5] Peruzzo, C., Simoni, L., Schrefler, B. A.: On Stepwise Advancement of Fractures and Pressure Oscillations in Saturated Porous Media. Engineering Fracture Mechanics (2019). https://doi.org/10.1016/J.ENGFRACMECH.2019.05.006

[6] Remij, E. W., Remmers, J. J. C., Huyghe, J. M., Smeulders, D. M. J.: An Investigation of the Step-Wise Propagation of a Mode-II Fracture in a Poroelastic Medium. Mechanics Research Communications (2017). https://doi.org/10.1016/J.MECHRESCOM.2016.03.001

[7] Cao, T. D., Hussain, F., Schrefler, B. A.: Porous Media Fracturing Dynamics: Stepwise Crack Advancement

and Fluid Pressure Oscillations. Journal of the Mechanics and Physics of Solids (2018). https://doi.org/10.1016/j.jmps.2017.10.014

Comment # 3
3) The comparison with data from Das et al. is interesting, and a bit more could be made of this. The main disagreement seem to be that the water-level change (i.e. the flux into the conduit) goes quite wrong in the model: much more water gets into the crack system than the model predicts. Interestingly, given the model is predicting the wrong amount of water in the system, the uplift prediction is quite good initially (although it also goes wrong at later times). The explanation about a pre-existing damage network seems plausible. I was surprised not to see more discussion about the possibility that the bedrock is not frozen to the ice: if the basal 'crack' or conduit can spread without cracking (a zero-fracture-toughness limit) then presumably the crack would spread further and allow more water in, without necessarily increasing the localised uplift (because the water has spread laterally further). It would, presumably, be straightforward to consider simulations with different fracture toughnesses for the ice-bedrock interface - and this seems valuable anyway, because we don't really know what that value should be. It also seems likely that the 2D planar assumption has quite significant errors as the spreading at the base continues, compared with an axisymmetric model, which is perhaps behind the later-time disagreement in the uplift (the geometric constraints are rather different for spreading as a circle compared to spreading as a line)? Again it would be useful to be able to compare the model predictions.

Indeed, the agreement between the performed simulations and the observations from Das et al. (2008) match quite well for the uplift, whereas the change in lake water-level has a larger mismatch. As pointed out by the reviewer, a potential source of mismatch could be the frozen base assumption, which is mentioned in the article at line 471: *"Additional assumptions for the numerical model are that the ice sheet is pristine (i.e. undamaged) and the ice-rock boundary is initially frozen, neither of which is strictly correct. Within the ice, pre-existing cracks, crevasses, and defects can link to the newly developed hydrofractures, which could significantly enhance the water inflow. Furthermore, fluid flow and movement at the ice-bed interface as it drains downglacier could influence the modelled fluid inflow and ice sheet uplift."*.

A major reason that this mismatch in drainage is observed is the conversion between observational and simulation data. Our 2D plane-strain simulations provide inflow rates per unit out-of-plane-width, whereas the observations report only the change in lake waterlevel, not inflow rates. To convert between these quantities, we have had to assume a crack length (dictating the conversion between 2D and 3D water inflow rates) and lake surface area (converting from 3D water inflow rate to changes in lake waterlevel). Both of these are taken from Das et al. (2008) and are assumed constant as only a single value is reported for these quantities. However, especially the lake surface area should change drastically over time, going from the reported surface area of 5.6 km$^2$ to zero when the lake is fully drained. Moreover, the results from Das et al. report that the height buoys used to record the lake height do not lower any further after 1.5 and 2.5 hours, indicating that the full lake is drained within this time-span. We do not have the relation between lake waterlevel and surface area, but if we were to include this, it would result in faster lake drainage rates over time (as the surface area decreases), making it feasible that a similar acceleration of drainage could be attained as was seen in the data from Das et. al. (2008). This is discussed from line 476 onwards: *"One final point of potential mismatch is the conversion between water volumes resulting from our simulations to the lake drainage height reported by Das et al. (2008) and conversely, from lake water level to volumes. We assume a simplified lake geometry with a constant area as water height decreases thus ignoring the effect of lake bathymetry. This simplification is therefore likely responsible for the model's underestimation of lake water level change particularly during the later stages of the lake drainage"*. Given the limitations with observations, it would be practically impossible to obtain inflow rates by measuring lake waterlevel.

Regarding whether using an axisymmetric assumption would be more appropriate for the side-ways spreading, we would like to note that Das et al. reported a 3.2 km long fracture, from which the fluid was likely spreading outwards at the base of the ice sheet. For axisymmetry to be valid, this 3.2 km length would need to be negligible compared to the region over which spreading occurs, such that it can be considered as a single point of inflow. After the 2 hours of our simulation, we only obtain a spreading of 1.6 km in each direction (or 600 m for a linear-elastic rheology), which indicates that considering the vertical crevasse as a single point source is definitely not a valid approximation. Of course a three-dimensional model would be most appropriate, capturing the uni-directional spreading near the centre of the crack while capturing the radial expansion near the edges, but unfortunately this is not feasible due to computational costs associated with this type of highly detailed simulations with our Matlab code. In ongoing work, we are developing a 3D parallel code and will likely revisit this. We have added the following statement to the paper, motivating our choice to use a plane-strain model: *"As our model predicts horizontal cracks*

*of* 1.6 km *in each direction, compared to the out-of-plane width of* $W_{\mathrm{oop}} = 3.2$ km*, assuming plane-strain conditions is reasonable for this case."*

Comment # 4

4) One of the aims of this work, as I understand it, is to highlight the role of the viscous rheology, and the fact that the interplay between viscous creep and elastic deformation can be very important in these processes. I think this point would be aided by a bit more analysis of the relative importance of the two modes of deformation. Specifically, the work compares the 'linearly elastic' model (just elasticity) with the 'viscoplastic model' (elasticity and viscous creep), but we don't really learn about how important viscous creep is relative to elasticity in the latter. i.e., one might be tempted to conclude that the role of viscous creep is dominant here, and that a third model of pure viscous creep (no elastic deformation at all) would do fine. It would be interesting to look at how much of a role elasticity is playing in the model, to be able to draw a clearer conclusion about how much the interplay of elasticity and creep is important here. That would be a helpful qualitative conclusion to draw from the work.

One of the main aims is indeed to highlight the importance of including viscous deformations. Historically, glaciology researchers used linear elastic fracture mechanics when considering whether cracks propagate through ice-sheets [1-3], and this has also been the assumption for published lake drainage hydro-fracture studies [4-5]. The standard argument is that fracture propagation occurs rapidly on timescales smaller than the Maxwell relaxation time, so it is reasonable to ignore viscous effects. While this is somewhat valid for vertical crevasse propagation, we find that the viscous deformations allow for a great uplift that enables transition from vertical crevasse propagation to horizontal basal crack propagation and sustains the flow of meltwater. Ignoring viscous deformations, leads to crack closure at the base of the vertical crevasse and shuts off any water flow into the horizontal crack. Thus, our work highlights that including viscous deformations in addition to elastic deformations significantly alters the propagation of the basal crack and lake drainage, even for the relatively short time-scales (minutes to hours) over which the crevasse propagation occurs.

Even though the viscous deformations provide a larger contribution to the total strain, the elastic processes still play an important role within our cohesive fracture model. Particularly, the elastic stresses are used to define the fracture propagation criterion and drive the dissipation through viscous deformations. As a result, we cannot simply "disable" elasticity in our model to see what would happen in a purely viscous setting, as this would require switching from a solid-mechanics-based framework (described through displacements and strains) to a fluid-mechanics-based framework (described through velocities, pressure and strain rates), where cracks nucleate based on the pressure or strain rates instead of considering the normal stress components.

[1] Van Der Veen, C. J.: Fracture Mechanics Approach to Penetration of Surface Crevasses on Glaciers. Cold Regions Science and Technology (1998). https://doi.org/10.1016/S0165-232X(97)00022-0
[2] Bassis, J. N., Walker, C. C.: Upper and Lower Limits on the Stability of Calving Glaciers from the Yield Strength Envelope of Ice. Proceedings of the Royal Society A: Mathematical, Physical and Engineering Sciences (2012). https://doi.org/10.1098/RSPA.2011.0422
[3] Lipovsky, B. P.: Ice Shelf Rift Propagation: Stability, Three-Dimensional Effects, and the Role of Marginal Weakening. The Cryosphere (2020). https://doi.org/10.5194/tc-14-1673-2020
[4] Tsai, V. C., Rice, J. R.: A model for turbulent hydraulic fracture and application to crack propagation at glacier beds. Journal of Geophysical Research: Earth Surface (2010). https://doi.org/10.1029/2009JF001474
[5] Tsai, V. C., Rice, J. R.: Modeling Turbulent Hydraulic Fracture Near a Free Surface. Journal of Applied Mechanics (2012). https://doi.org/10.1115/1.4005879.

Comment # TC1

TC1) Throughout: the non-linear visco-elastic formation is throughout referred to as a 'viscoplastic' law. I know this terminology is sometimes used to describe Glen's flow law, but it isn't strictly correct, and anyone from a non-Newtonian fluids / rheology background would be confused by its usage here. They would traditionally Glen's flow law as a shear thinning viscous rheology - and the model used here would be a viscoelastic model: the ice comprises elastic deformation (recoverable) and non-linear viscous deformation (non-recoverable, and given by a shear-thinning model). A 'viscoplastic' model would typically be taken to indicate that there is a plastic 'yield' stress, below which there is no non-recoverable deformation; at which the material deforms plastically; and above which the material flows viscously (see e.g. much literature on visco-elasto-plastic models). I would favour not describing the formulation here as 'viscoplastic'.

Indeed, as pointed out by the reviewer, both "viscoplastic" and "viscoelastic" are used within literature to denote the combined viscous Glen's law and linear-elastic deformation model. The definitions are different based on the community - solid versus fluid mechanics or experiment versus theoretical mechanics. For example in computational solid mechanics, a viscoplastic rheology describes a behaviour where deformations are irreversible and time-dependent, with the prefix visco- used to indicate its time-dependence). For example, some models are described as viscoelastic-viscoplastic, meaning that both reversible and irreversible deformations exhibit time or rate dependence. In contrast, in experimental solid mechanics, a 1D viscoelastic model is simply a viscous dashpot combined (either in series or parallel) with an elastic spring. Although the computational solid mechanics definition which we used is more robust for defining material behaviour in 3D, we are sympathetic to the reviewer's concern that this will confuse the non-Newtonian fluid mechanics community. We have now updated the usage of viscoplastic throughout our manuscript to now read "viscoelastic", in response to both reviewers' comments. This is anyway mostly semantics at this point. We have also added a footnote to clarify this usage for the broader mechanics community: *"The term "viscoelastic" is used to refer to the combination of reversible elastic deformations and irreversible viscous/plastic deformations resulting from the use of Hooke's and Glen's laws. While this terminology is more common in the non-Newtonian fluids/rheology and glaciology community, in the solid mechanics community these models are referred to as viscoplastic models, namely the Norton-Hoff and Bingham-Maxwell models."*

Comment # TC2

TC2) p.6 Figure 3: needs to say this is the range of Maxwell times - the caption just says 'range of time-scales' which could mean anything.

Added Maxwell into the captions of Figures 3 and 4 to indicate that Eq. 4 indeed gives the Maxwell time scale: *"Range of Maxwell time-scales following from Eq. 4"*.

Comment # TC3

TC3) p.6 equation (3) has some weird typesetting on the third line (missing equals?)

This equation (and all others) have been type-set with the line endings compatible with a two-column paper, such that the final version does not exceed the allowed line length. It is correct that this matrix multiplication spans two lines.

Comment # TC4

TC4) p.9 The authors choose a Manning-Strickler turbulent flow law following Tsai and Rice and others. Do any of the results have any appreciable dependence on this choice as opposed to other turbulent laws? (e.g. see opening of Dontsov 2016 J. Fluid Mech. 'Tip region of a hydraulic fracture driven by laminar-to-turbulent fluid flow' or final section of Hewitt et al. 2018 J. Fluid Mech. 'The influence of a poroelastic till on rapid subglacial flooding and cavity formation'). Those studies also give details of how to map to a laminar regime if the fluid velocity / crack dimensions become too small: could this be important at later times? (Particularly in the case where the linear-elastic model shuts off the water supply to the basal crack, which then slowly continues to spread - see point below.)

Indeed, the results do depend on the used flow law within the fracture. The rate of the water flowing downwards through the crevasse is one of the main factors describing the rate of crevasse propagation. The pressure inside the crevasse can only build up as long as fluid is transported towards the crack tip, and while part of this fluid is absorbed by the opening induced by viscous deformation (hence the crack propagates slower for the viscoelastic model compared to the linear-elastic), the dominant time-scale of the problem is dictated by the fluid flow. In the case of a linear-elastic material, the fluid flow provides the only time-dependent terms (except for the inertia term, which operates on much smaller time-scales), thus directly it controls the rate of fracture propagation.

If a laminar flow model is used, the water flows much faster through the crevasse, its propagation would be faster compared to turbulent flow model. To provide an example of the impact of flow model, the results from Figs. 2 and 3 use a laminar fluid flow model, with the fluid flux given by the cubic law as $q = -h^3/12\mu \; \partial p_d/\partial \xi$ (a standard model for simulating hydro-fracture, based on the analytic solution for pressure-driven flow between two flat plates). The main difference is indeed the much higher flow rates under the laminar flow model, causing the crevasse to reach the base within minutes, and propagate side-ways rapidly. As this flow rate greatly exceeds the volume created due to viscous deformations, no breaks in propagation (where the viscous opening rate perfectly balances the fluid inflow rate) are observed, and the sole "pause" in the crack propagation is observed due to the crevasse approaching the end of the domain. We have checked that, based on the flow rates observed in our simulations, the fluid flow is definitely in the turbulent regime with a Reynolds number of $Re \approx 3 \cdot 10^5$ (see text

[Figure]

Figure 2: Fluid flow rate and lake drainage when a laminar fluid flow model is used. Please note that the results shown are from preliminary results, and not as well-verified as the results from the main paper. Fluid flow rate is reduced after $\approx 20$ minutes as the horizontal crack approaches the domain boundaries.

[Figure]

Figure 3: rate of crack propagation when a laminar fluid flow model is used. Please note that the results shown are from preliminary results, and not as well-verified as the results from the main paper. Results after $\approx 20$ minutes are heavily influenced by the domain boundaries.

below Eq. 7 in the paper, lines 231-235), and thus the results presented in the above discussion using a laminar flow model are definitely unrealistic.

We have not performed sensitivity studies using different fracture flow models, and the parameters used for the friction factor and wall roughness are directly taken from Tsai and Rice [1,2] without altering these values to better fit the observed lake drainage. Our view is that it is beyond the scope of the presented paper to alter every possible modelling parameter and assumption to obtain the best possible match to observed results, and as such we only present results using a single flow model leading to the conclusion that viscous deformations are important when considering hydro-fracture. However, all codes used are provided via the data availability statement (and the laminar flow model is already implemented), such that others can build upon this work and investigate the importance of parameters aligning with their interests.

[1] Tsai, V. C. and Rice, J. R.: A model for turbulent hydraulic fracture and application to crack propagation at glacier beds, Journal of Geophys- ical Research: Earth Surface, 115, 3007, `https://doi.org/10.1029/2009JF001474`, 2010.
[2] Tsai, V. C. and Rice, J. R.: Modeling Turbulent Hydraulic Fracture Near a Free Surface, Journal of Applied Mechanics, 79,720 `https://doi.org/10.1115/1.4005879`, 2012

Comment # TC5

TC5) p.10 Equation (12) does not make sense. The condition is supposed to be a fixed pressure (i.e. at the base of the lake), so what is this 'penalty' term added to (12), and how is the parameter chosen (the text says it is chosen 'to be large enough to enforce this inflow condition', which isn't clear: it is part of the inflow condition, so the value you pick for it will change that condition.) In addition, if the problem is being driven by a fixed pressure condition at the start of the fracture, than I don't see what the variable p is chosen to be at that location - it should surely be $p = p_{\text{ext}}$, but that seems inconsistent with equation (12). This means I don't really know what is being imposed on the pressure in the model, which is awkward when trying to interpret the results of the model.

Indeed, the boundary condition at this inlet is $p = p_{\text{ext}}$. Mathematically, there are two manners in which this boundary condition can be included within a finite element scheme: It can either be directly substituted within the system of equations being solved (eliminating a single pressure degree of freedom to be solved for, and moving related terms to the external force vector), or an additional equation enforcing $p_{\text{ext}} - p = 0$ can be used (keeping the inlet pressure as degree of freedom to solve for, but prescribing what its value should be). The penalty method is a commonly used to add constraints following this second method, with its main advantage being that it allows constraints to be imposed, while also allowing the forces (or in the case of pressures, fluid fluxes) that are resultant of this constraint to be evaluated. Within this method, a forcing term $q = k_p(p_{\text{ext}} - p)$ is added, with $k_p$ being much larger than any other term for this degree of freedom within the system of equations. As a result, it effectively substitutes the mass balance that describes the pressure at the inlet with $k_p(p_{\text{ext}} - p) \approx 0$, such that $p_{\text{ext}} \approx p$. Indeed, as pointed out by the reviewer, this does not enforce the pressure to be exactly the boundary pressure, but with the high value used for $k_p$ this is extremely close to the pressure. For our values used, $|p/p_{\text{ext}} - 1| < 10^{-5}$, producing a negligible difference.

To clarify that this boundary condition is equivalent, and the way it is imposed is a purely numerical consideration, we have added the following to the paper: *"We elect to enforce this pressure boundary condition in a weak sense through a penalty approach, such that the fluid influx at the inlet can be easily recorded. It should be noted, however, that using Eq. (12) with a high value for $k_p$ is equivalent to directly enforcing this pressure, and we have verified that the pressure at the inlet is approximately equal to the applied $p_{ext}$."*

Comment # TC6

TC6) p.11 equation (15) seems to have the wrong units for a heat flux (is there a missing $c_p$?)

The units for Eq. (15) are correct. This equation provides the frictional heating, and is thus dependent on the energy dissipated at the walls, not on the heat capacity of the ice or the water. The units used in this equation are:

$$q \text{ (fluid flux, } \int v \, dh) : \qquad \text{m}^2\text{s}^{-1}$$
$$\frac{\partial p}{\partial \xi} \text{ (pressure gradient)} : \qquad \text{Pa m}^{-1} \tag{1}$$

Thus giving the thermal flux as energy per unit area per second:

$$j \text{ (thermal influx)} : \qquad \text{m}^2\text{s}^{-1} \cdot \text{Pa m}^{-1} = \text{Pa ms}^{-1} = \text{Jm}^{-2}\text{s}^{-1} \tag{2}$$

[Figure]

Figure 4: Comparison between Figure 6 from the paper (left), showing the pressure as a single line, and the crack-filled version (right), showing the water pressure filling the complete crack opening.

using the definition of the energy $J = N\,m = kg\,m^2\,s^{-2} = Pa\,m^3$

Comment # TC7

TC7) Equation (16-18) I don't think eta is defined here. More importantly, the expression in (18) is obviously wrong and seems to have been lifted from elsewhere - the factors of pi and the power of $(3/2)$ rather than $(1/2)$ on the $(t\text{-}t\_0)$ indicate this is the corresponding expression for an axisymmetric fracture, not a plane fracture. Equation (19) has the same issue - and so this error will propagate into all of the numerical results.

The symbol $\eta$ is defined as the normal coordinate in the fracture-local coordinate system, see Figure 2 and the first paragraph of Section 2.2.1. A clarification has been added to the paper regarding this below Eq. 16: *", and using the surface-normal coordinate $\eta$."*

Regarding the occurrence of $\pi$ within Eq. (18), this is not related to the axisymmetric assumption. Instead, it comes from the analytic solution for the temperature profile containing the complementary error function erfc, as its derivative contains $1/\sqrt{\pi}$ term. The provided solution is valid for flat walls of planar fracture (following the well-known Stokes' first problem/Rayleigh's problem), and follows the derivation given in [1], see also [2-4], all of which consider flat plates and obtain this factor $\pi^{-1/2}$ upon calculating the wall shear stress (mathematically equivalent to the thermal flux).

[1] White, F. M.: Viscous Fluid Flow, McGraw-Hill, New-York, 3 edn., 2006. Specifically: Section 3-5: Unsteady flows with moving boundaries, pages 129-130
[2] Kharchandy, S., (2018). Exact Solution for Unsteady Flow of Viscous Incompressible Fluid Over a Suddenly Accelerated Flat Plate (Stokes' First Problem) Using Laplace Transforms. International Journal of Engineering & Technology, 7(3.6), 267-269. `https://doi.org/10.14419/ijet.v7i3.6.15000`
[3] Hu SP, Fan CM, Chen CW, Young DL. Method of Fundamental Solutions for Stokes' First and Second Problems. Journal of Mechanics. 2005;21(1):25-31. `https://doi.org/10.1017/S1727719100000514`
[4] Wikipedia: Rayleigh problem `https://en.wikipedia.org/wiki/Rayleigh_problem`

Comment # TC8

TC8) p.15 Figure 6 looks a bit odd: it would be much more clear if the colours filled the gaps, rather than being a line in the centre of the gap.

We are unsure why representing the pressure along the cracked interface as a coloured line is not clear. From the viewpoint of representing how this pressure is captured within simulations, it is more accurate to represent the pressure as a one-dimensional line overlaid on the 2D ice-sheet (as within simulations the crack is considered a single line interface along which the pressures are defined and across which displacement are discontinuous). If we were to show the fluid pressure to fill the complete interior of the crack, it might create the impression that the water pressure is solved in two-dimensions (e.g. by solving Navier-Stokes equations). Additionally, if we were to only show the fluid pressure when the crack has opened, the horizontal crack length for the linear-elastic case would no longer be visible (see Fig. 4). Therefore, we feel that the original Figure 6 is clearer.

Comment # TC9

TC9)  p.16 Line 370. The statement that the crevasse closure stops the propagation of the horizontal crack is inconsistent with figure 7, where the crack continues to spread slowly after the closure (i.e. the blue line keeps rising after about t=45min, when the gap closes). It is unclear to me why the lateral fracture continues to spread, even though the entry has closed over: if it has enough pressure to keep spreading at the tip, why does it not have enough to force the original opening back open?

Because water expands upon freezing to ice, there is a very slight increase in pressure, which continues to pressurise the crack (see Figure 10 in the revised manuscript, showing the continuing freezing). We have updated the statement that the crack propagation stops as the fluid inflow stops to clarify that it becomes negligible instead of fully stops, with the only increase in crack length due to freezing. *"This prevents any further fluid from entering the crevasse, causing the only slight increases in pressure due to the freezing of fracture walls, and thus eventually stopping the propagation of the horizontal crack at the ice-bedrock interface."*

---

## Author Comment (AC2)

**Authors' response to reviewers report**

**Response to Reviewer 2**

This is an interesting model study of an important process. The model represents an advance on previous treatments of hydrofracture, and has yielded some interesting new insights into the relative importance of thermal and deformational processes in controlling rates of hydrogfracture propagation and supraglacial lake drainage. The anonymous referee has raised a number of technical points regarding the paper, and I shall not repeat them here.
   ...
   These comments aside, I like this paper, and commend the authors on some interesting work. With the addition of a bit more context (and subject to the technical issues raised by the other referee), I recommend publication.

We thank the reviewer for his comments and positive recommendation. Below, have addressed the points raised by the anonymous reviewer 1, in our the response above. Changes to the manuscript are in blue within the manuscript.

Comment # 1
1) With regard to terminology, I agree that the term 'visco-plastic' is not appropriate and should be changed to "viscous" or "non-linear viscous" throughout. It is also unnecessary to refer to the "so-called Glen's flow law" (line 126). Although some have used other names for this flow law from time to time, "Glen's Flow Law" is in very widespread use, so the "so-called" is superfluous.

Based on the comments of reviewer 1 (see reviewer 1, response #TC1) , we have updated our terminology throughout the manuscript to refer as the model describing linear-elastic and viscous deformation as visco-elastic model and have changed the term "visco-plastic" to "viscous", to avoid misunderstanding. We have also updated line 126 to read as simply *"Glen's flow law"*. We do note this terminology is different between computational solid mechanics and non-Newtonian fluid mechanics communities, and also experimental versus theoretical modelling communities.

Comment # 2
2) My main comments concern model formulation and how it may relate to reality. The 2D plane geometry seems to me to be perfectly adequate to explore the issues of interest, and there is perhaps no need to add experiments with an axisymmetric geometry (full 3D would of course be much better still).

Please see our response to reviewer 1, comments #1 and #3, where we justify the use of the 2D plane geometry. We have added the following to the manuscript:
   *"Using an axisymmetric representation would be appropriate for short crevasses and conduits with lengths much shorter compared to the length of the horizontal/radial basal crack, whereas the plane-strain representation used here is suitable for when the surface crevasse spans longer distances."*
   *"Of course a 3D model able to capture both these phenomena would be ideal, but it would be computationally too expensive, so we utilize the 2D plane strain approximation, focusing on the propagation of fractures driven by stresses and only consider the lateral melting of the fracture faces."*

Comment # 3
3) More serious for present purposes is the assumption of a frozen bed. In Greenland, the bed is temperate below most of the ablation zone, and water is certainly present at the ice-bed interface. This has three important implications for any attempts to compare the model results with reality. First, it is not necessary to form a fracture along the bed, simply to lift it.

We acknowledge that the ice-bed interface is not frozen over much of the Greenland Ice Sheet's ablation area, particularly in the locations where supraglacial lakes drain via hydrofracture (and this is recognised in the text on lines 95-97). We address the frozen-bed assumption in more detail below and in Section 2.1.2 in the text. We do recognise that this assumption does require more explanation as to not confuse the reader. We have therefore updated the text by adding a statement on line 95 directing the reader to the full discussion of this assumption at the first mention of the frozen-bed assumption. In this statement we also reiterate the fact that the ice-bed interface is this region is not frozen. We address the reviewer with the full reasoning for this model assumption below.
   The crack propagates when the vertical stresses exceed the tensile strength of the cohesive interface, such that the difference in the stress required for horizontal cracking/uplifting to occur $\sigma_{yy}^{crack}$ and the stress without any

crack must exceed the tensile strength $\sigma_{yy}^0$ and the weight of the ice, that is:

$$\sigma_{yy}^{crack} - \sigma_{yy}^0 > f_t + \rho_i gH. \tag{1}$$

The magnitude of these terms for the 980 m thick ice-sheet considered are $f_t = 0.14$ MPa and $\rho_i gH = 8.74$ MPa, with the pressure within the fracture needing to be high enough to overcome the combination of these. If instead the base was not frozen, the tensile strength needed to be exceeded would be zero, but the majority of the stress that needs to be overcome from the gravity contribution would still exist, with comparable water pressure at the base being required for the uplifting. The effect of this would be that the crack starts propagating slightly sooner, with the pressure within the crevasse being slightly lower at the moment the horizontal propagation commences. However, from that point onwards, the rate of propagation is governed by the volume created due to vertical displacement of the ice-sheet, and the rate at which water flows down through the vertical crevasse; the lack of any tensile strength would not create a significant difference. To clarify that the majority of the resistance to horizontal crack propagation and uplifting at the base comes from the weight of the ice-sheet, we have added the following to the paper:

*"The crack propagates once the stress within the ice, normal to the prescribed crack direction, exceeds the tensile strength, $\sigma_{yy} > f_t$ for the horizontal crack and $\sigma_{xx} > f_t$ for propagation of the vertical crevasse. As the weight of the ice induces compressive (negative) stresses within the ice-sheet, the total change in stress needed to propagate the crack is therefore given by $(\boldsymbol{\sigma} - \boldsymbol{\sigma}_0) \cdot \mathbf{n} > f_t + \rho_i g(H - y)$. For the horizontal crack, this implies that even through a frozen rock-ice interface is assumed with a tensile strength, the majority of the stresses that need to be overcome are a result of the weight of the ice and not the tensile strength."*

Of course, a second difference of the bed not being frozen is the possibility of sliding to occur pre-fracture, enhancing the vertical crevasse opening by the two fracture faces "sliding" apart, which is addressed as:

*"It should be noted that assuming the bed to be frozen has implications for the downward crevasse propagation and crevasse opening width after the vertical crevasse reaches the base, which is only driven by the elastic and viscous deformations. In contrast, were frictional sliding be allowed at the glacier bed even before the onset of horizontal crack and uplift, an additional opening width would be created due to the two "sides" of the ice-sheet sliding apart. The effect of the glacier sliding induced crevasse widening would be significant if the basal friction is weak."*

It should be noted that, if a non-frozen bed was assumed, this free-slip enhanced crevasse opening would be significantly impacted by the plane-strain assumption. In a three-dimensional setting, the ice ahead and behind the surface crevasse would still connect the ice, limiting how much the sides can slide apart. In contrast, under plane-strain, representing an infinitely long crevasse in the out of plane direction, the two sides of the ice-sheet would be completely disconnected by the crevasse, thus allowing for a much larger sliding. As such, assuming free-slip (or frictional) basal conditions would not produce realistic results for a temperate base either without the addition of an additional body force representing the three-dimensionality of the crevasse (or performing full 3D simulations).

Comment # 4

4) Second, interactions with the basal drainage system will have potentially large influence on the fate of water arriving via a hydrofracture. Third, enhanced slip at the bed will not necessarily be confined to local 'blisters' as implied by the model. Each of these three considerations mean that real-world drainage events can play out in ways not simulated in the model. The points about basal drainage and basal slip are especially important, as the authors invoke concern about sea level rise as a justification of their work. In the frozen-bed scenario assumed in the model, any slip will be local and bounded by the surrounding frozen bed; in the real world, slip is less constrained, and can either be enhanced (if the extra water increases areas of ice-bed separation) or reduced (if the extra water encourages development of an efficient conduit). Both of these effects have been described as consequences of surface to bed drainage in Greenland, and show that interactions between hydrofractures and basal drainage are of great importance.

I am not suggesting that the authors include a basal drainage component in their model. Science often proceeds incrementally, and it is unreasonable to expect that this study should solve the complete problem. I am suggesting though, that the authors more fully acknowledge the complexity of the full problem, and more carefully identify which issues their paper has explored and which ones remain unsolved.

We address the reviewer's comments about the frozen-bed assumption in Comment #3 above and here focus on the broader points raised in this comment regarding the basal drainage component and basal slip.

We would like to first thank the reviewer for their clear explanation of the complexity of the interactions between surface meltwater induced hydrofracture and basal drainage interaction. We have addressed the reviewers concerns regarding the distinction of our model's results versus future model applications by revising Section 4.4 entitled

"Future of the Greenland Ice Sheet" in the main text. In this revision we make clear the scope of this model's application (i.e., the first hour of the lake drainage event and exploring local conditions only) and how these results can be coupled with other models or observations to investigate lake drainage induced changes to the subglacial drainage system and overall impacts on ice dynamics. Specifically, we do not currently include interactions with the subglacial drainage system because rapid lake drainage via hydrofracture should quickly overwhelm any existing subglacial drainage system. Regarding interactions with the subglacial drainage system after the formation of the local blister, this can be investigated by coupling our model with a subglacial hydrology model to include a drainage term to remove water from the blister. However, because this current paper focuses on the first 90 minutes of lake drainage when the blister is forming we do not include that here and instead discuss how to incorporate basal drainage system interactions in Section 4.4.

The added text to Section 4.4 reads: *"While the presented model is restricted in scope to only solve for a single hydro-fracture and the resulting uplift, results can be coupled with data sets and other process based models to fully investigate ice dynamic response to rapid supraglacial lake drainages. For example, crevasse formation models (e.g., Hoffman et al., 2018) or remote sensing data can constrain the timing of supraglacial lake drainages with the later also defining pre-drainage lake conditions. Results from our subsequent model runs can be fed back into large-scale low-resolution glaciological models, or smaller floodwave propagation models (e.g., Lai et al., 2021) to inform changes in basal conditions both locally and downstream along the resulting subglacial floodwave."* ... *"While our model does not incorporate a basal drainage component nor simulate the subglacial floodwave produced by the lake drainage event, these complex processes can be investigated in the future by coupling our model with a subglacial hydrology model to comprehensively assess the ice-dynamic and hydrological consequences of rapid lake drainages. "*

[1] Dow, C. F. et al. (2015). Modeling of subglacial hydrological development following rapid supraglacial lake drainage. J. Geophys. Res. Earth Surf., 120, 1127–1147. https://doi.org/10.1002/2014JF003333.

Comment # 5
5) It is also worth highlighting that the model also requires the location of initial crack to be specified, a limitation that will need to be overcome before exploring the kind of "flood wave" scenario the authors invoke in lines 495 and forward.

Indeed, the location of initial cracks and the expected crack path need to be known and defined at the onset of the simulation, which is a weakness of the described model, and more broadly, a weakness of interface element-type representations of fracture processes. While this pre-determined crack path is already shown in Fig. 2, we have added the text below to the start of the methods section to make clear this limitation. We also discuss the use of satellite imagery or another process-scale model to define the locations and timing of supraglacial lake drainages to inform the application of our model.

*"It is noteworthy that the path along which new interface elements are inserted is pre-determined, allowing the crevasse to only propagate straight down, and then splitting into two basal cracks that can only propagate sideways in a straight line. While the pre-determination of crack path and insertion of cohesive interface elements only between continuum finite elements are limitations of our numerical approach, it is reasonably realistic given the 2D idealisation of the rectangular glacier domain. The requirement of the crack path to be known* a priori *restricts the nucleation of crevasses elsewhere in the domain. Also, we do not model the surface hydrology associated with the formation of supra-glacial lakes, but rather assume a pre-existing lake with known depth that intersects with this initial crack."*

Due to this limitation, the model indeed can not be used to study the nucleation of water-driven crevasses throughout an ice-sheet, and capture how the water transport to the glacial bed impacts the glacial movement. However, by informing the model with the data provided from large-scale glaciological models (providing strain rates, geometry, and surface water-levels), we may be able to produce high resolution predictions of the rate at which a single crevasse will transfer large amounts of melt-water to the bed; this meltwater flux can then be fed back into large-scale models to predict more long-term behaviour. We have clarified this potential usage of this model, indicating that it does not capture the full problem but only the part requiring high-fidelity simulations, in the discussion: *"While the presented model is restricted in scope to only solve for a single hydro-fracture and the resulting uplift, results can be coupled with data sets and other process based models to fully investigate ice dynamic response to rapid supraglacial lake drainages. For example, crevasse formation models (e.g., Hoffman et al., 2018) or remote sensing data can constrain the timing of supraglacial lake drainages with the later also defining pre-drainage lake conditions. Results from our subsequent model runs can be fed back into large-scale low-resolution glaciological models, or smaller floodwave propagation models (e.g., Lai et al., 2021) to inform changes in basal conditions both locally and downstream along the resulting subglacial floodwave."*

---

## Author Response (AR2)

Dear Editor,

Thank you for accepting the article with these minor modifications. In response to the reviewer comments, we have changed:

1) The reviewer is correct, and eq. (18) indeed contained typographic errors. These have been corrected in the updated manuscript, and we apologise to the reviewer for initially dismissing their comment regarding this equation being incorrect (the factor π is still not related to axisymmetry, but the second part of the reviewer's first comment should have warranted more carefully checking on our part). We have changed this equation to now have a factor 2 in front (please note the minus sign included in the -2k term gets cancelled by the sign of the $-2/\sqrt{\pi}$ term), and use the exponent ½ for the time. We have also expanded upon the derivation in the paper to make it easier to follow. Please note that this factor 2 cancels out in Eq. 20, due to Eq. 10 also containing a factor 2, which was correctly incorporated in the original manuscript. Furthermore, all simulation results that are reported in the original manuscript have used the corrected formulation, see the publicly available github page for this code to indicate that the results have always been obtained using a factor 2, and an exponent of ½ (e.g., see line 434 for this thermal flux in the file https://github.com/T-Hageman/MATLAB_IceHydroFrac/blob/main/Models/%40FractureFluid/FractureFluid.m)

2) We thank the reviewer for this suggestion, and will take it into consideration for future work. To clarify that we indeed use a single flow model, but that it theoretically would be possible to switch between laminar and turbulent models, we have added a couple of sentences to section 2.2.1: "We note that, while this fluid flux is indeed turbulent in the majority of the crevasse, it is unlikely to be close to the crack tips. For simplicity, we use this turbulent relation throughout the crevasse, however, slightly more realistic results could potentially be obtained by dynamically switching between laminar and turbulent flow models based on the current fluid flux."

We hope that these additions have satisfied the Editor and Reviewers,

Kind regards,